# Modelling EWS::FLI1 protein fluctuations reveal determinants of tumor plasticity in Ewing sarcoma

Veveeyan Suresh [ID] [1,8], Christoph Hafemeister [ID] [1,8], Andri Konstantinou [ID] [2,3], Sarah Grissenberger[1], Caterina Sturtzel[1], Martha M Zylka [ID] [1], Florencia Cidre-Aranaz [ID] [4], Andrea Wenninger-Weinzierl[1], Karla Queiroz[5], Dorota Kurek[5], Martin Distel [ID] [1,7], Anna Obenauf [ID] [2], Thomas G P Grünewald [ID] [4], Florian Halbritter [ID] [1], Heinrich Kovar [ID] [1,6,9 ✉] & Valerie Fock [ID] [1,9 ✉]

## Abstract

**Tumor cell plasticity drives metastasis and therapy resistance, yet its regulation by oncoprotein dosage dynamics remains poorly understood. In Ewing sarcoma (EwS), variations in EWS::FLI1 (EF) fusion oncoprotein activity have been associated with epithelial-mesenchymal plasticity (EMP). Using degron technology, we precisely modulated endogenous EF in EwS cells and linked phenotypic states to distinct oncoprotein dosages. Strikingly, modest EF depletion promoted a pro-metastatic phenotype that diminished upon near-complete EF loss, revealing a paradoxical effect of submaximal EF inhibition. Nascent RNA-sequencing uncovered distinct gene clusters with heterogenous transcriptional responses to graded EF loss. Genes most sensitive to subtle EF depletion harbored GGAA microsatellites within EF-bound enhancers, while chromatin profiling uncovered candidate cofactors regulating EF-repressed EMP programs. Transient EF depletion followed by rapid restoration, modelling oncoprotein fluctuations, caused persistent dysregulation of genes functionally linked to enhanced extravasation and metastatic burden in preclinical models. This study highlights the therapeutic challenge of incomplete EF elimination, serving a paradigm in which oncoprotein dosage dynamics act as non-genetic drivers of disease progression and reveal novel vulnerabilities of advanced disease.**

**Keywords** Epithelial Mesenchymal Plasticity; Ewing Sarcoma; EWS::FLI1; Metastasis; Oncogene Fluctuation
**Subject Categories** Cancer; Musculoskeletal System

See also: M Castets et al

## Introduction

Ewing sarcoma (EwS) is an aggressive bone and soft tissue cancer predominantly affecting adolescents. Despite intensive multimodal therapy, prognosis remains poor for patients with relapsed or metastatic disease (Zöllner et al, 2021). This highlights the urgent need for the development of more effective treatment strategies. Among potential therapeutic targets, the EWS::FLI1 fusion protein (EF) expressed in 85% of EwS cases has emerged as the central oncogenic driver (Delattre et al, 1992; Uren and Toretsky, 2005). EF functions as an aberrant ETS family transcription factor that binds to consensus sites containing a GGAA core motif in promoters and enhancers, and exhibits dual activity as a transcriptional activator and repressor (Bilke et al, 2013; Riggi et al, 2014). By rewiring the epigenome, EF can turn bound GGAA microsatellites into de novo enhancers activating nearby genes in an EwS-specific manner (Gangwal et al, 2008; Guillon et al, 2009; Riggi et al, 2014). EF-driven gene repression occurs either directly through interactions with co-repressors, or indirectly by activating other transcriptional repressors and recruiting repressive epigenetic complexes (Niedan et al, 2014; Owen et al, 2008; Sankar et al, 2014; Theisen et al, 2021). This leads to the downregulation of genes critical for mesenchymal identity and affects tumor cell plasticity (Tirode et al, 2007; Tomazou et al, 2015).

EwS is characterized by quiet genomes with few recurrent mutations (Brohl et al, 2014; Crompton et al, 2014; Lawrence et al, 2013; Tirode et al, 2014) but exhibits a significant degree of epigenetic plasticity and intra-tumoral heterogeneity (Apfelbaum et al, 2022a). Supporting EF role in driving plasticity, several studies have linked the existence of multiple tumor cell phenotypes in EwS to variations in EF levels and activity (Aynaud et al, 2020; Franzetti et al, 2017; Khoogar et al, 2022). Single-cell RNA-profiling of patient-derived cell lines and xenografts, and immunohistochemistry on primary tumors identified a subpopulation of ~1–2% dormant-like cells associated with low EF expression (Franzetti et al, 2017; Khoogar et al, 2022). EF knockdown models suggest

[1]St. Anna Children's Cancer Research Institute (CCRI), Vienna 1090, Austria. [2]Research Institute of Molecular Pathology, Vienna Biocenter, Vienna 1030, Austria. [3]Vienna Biocenter PhD Program, a Doctoral School of the University of Vienna and the Medical University of Vienna, Vienna, Austria. [4]Hopp-Children's Cancer Center (KiTZ), Heidelberg 69120, Germany. [5]Mimetas, DH Oegstgeest, Oegstgeest 2342, The Netherlands. [6]Department of Pediatrics, Medical University of Vienna, Vienna 1090, Austria. [7]Present address: Division of Pediatric Hematology and Oncology, Intermountain Primary Children's Hospital, Huntsman Cancer Institute, Spencer Fox Eccles School of Medicine at the University of Utah, Salt Lake City 84113 UT, USA. [8]These authors contributed equally: Veveeyan Suresh, Christoph Hafemeister. [9]These authors jointly supervised this work: Heinrich Kovar, Valerie Fock. ✉E-mail: heinrich.kovar@ccri.at; valerie.fock@outlook.com

that high EF levels are linked to a proliferative phenotype, whereas cells with low EF expression display mesenchymal features and an increased propensity for migration and metastasis (Bierbaumer et al, 2021; Chaturvedi et al, 2014; Chaturvedi et al, 2012; Franzetti et al, 2017; Katschnig et al, 2017). Several mechanisms are known to modulate EF activity, including competitive binding at shared chromatin binding sites with transcription factors such as ETV6 and HOXD13,(Apfelbaum et al, 2022b; Gao et al, 2023; Lu et al, 2023) as well as loss of STAG2, which is frequently subclonal and associated with poor prognosis (Adane et al, 2021; Surdez et al, 2021). Additionally, regulation on the transcriptional and RNA/protein stability levels directly affects EF expression (Garcia-Dominguez et al, 2021; Gierisch et al, 2019; Keskin et al, 2020; Kim et al, 2022; Riggi et al, 2010; Seong et al, 2021; Su et al, 2021; Wang et al, 2023a).

Fluctuations in EF levels may occur stochastically or in response to microenvironmental cues (Apfelbaum et al, 2022a; Bai et al, 2019; Gupta et al, 2011). Factors such as hypoxia, serum starvation, and signaling through Rho or Hippo pathways can impact EF expression and activity (Bierbaumer et al, 2021; Franzetti et al, 2017; Katschnig et al, 2017). However, a comprehensive model that captures the full spectrum of EF expression and activity levels in EwS has been lacking. This knowledge gap hinders the development of novel therapeutic strategies aimed at overcoming the inherent metabolic and epithelial-to-mesenchymal plasticity (EMP) of EwS cells. We engineered an EwS cell line to enable precise, tunable and reversible modulation of EF dosage using the dTAG system (Nabet et al, 2018). This allowed us to establish a gradient of distinct EF dosages to describe the short-term and long-term phenotypic and molecular consequences of EF fluctuations at defined amplitudes and identify a set of persistently dysregulated genes involved in EMP. Our results highlight the crucial role of EF dosage dynamics in EwS cell plasticity and disease progression with important therapeutic consequences.

# Results

## Degron knock-in for tunable EF expression from the endogenous gene locus

To precisely modulate endogenous EF dosage in EwS cells, we made use of the dTAG system, which allows rapid and tunable protein degradation (Nabet et al, 2018). We employed a CRISPR genome editing approach to tag endogenous EF in A673 EwS cells with the fluorescent protein mNeonGreen (mNG), serving as a quantitative proxy for EF levels. Additionally, we introduced an FKBP12$^{F36V}$ tag, facilitating targeted degradation of the fusion protein upon treatment with the heterobifunctional von Hippel-Lindau (VHL)-binding degrader molecule, dTAG$^V$-1 (Fig. 1A). Using FACS-mediated single-cell sorting followed by clonal expansion, we obtained clones with biallelic knock-in of the mNG-HA-FKBP12$^{F36V}$ tag at the C-terminal end of the FLI1 coding region (EF-dTAG clones) (Fig. EV1A). Consistent with a recent preprint (McGinnis et al, 2024), targeting the EF locus in other EwS cell lines (TC71, STA-ET-7.2) resulted in duplication of the EF allele with only one copy successfully modified, indicating selective pressure against this EF manipulation (Appendix Fig. S1A). Focusing on A673 EF-dTAG clones A2.2 and B3.1, Western blot

analysis showed that the absolute levels of tagged EF were similar between the clones and comparable to their untagged wild-type counterpart (Fig. EV1B). Consistent with the absence of expression of the unrearranged FLI1 allele, no signal was obtained for wild-type FLI1 for both the parental cells and EF-dTAG clones (Fig. EV1B). CUT&RUN revealed similar genome-wide chromatin binding patterns between untagged and tagged EF in parental cells and A673 EF-dTAG clone A2.2, respectively (Fig. EV1C; Dataset EV1). RNA-seq analysis identified 564 and 302 differentially expressed genes (DEGs) in EF-dTAG clones A2.2 and B3.1, compared to parental A673 cells, respectively, with an enrichment of genes linked to epithelial-mesenchymal transition (EMT) (Fig. EV1D; Dataset EV2). A similar number of DEGs was detected when comparing individual untagged A673 single-cell clones with the parental bulk population, suggesting that transcriptional differences arise from clonal heterogeneity or the single-cell cloning process (Dataset EV2). The EF-dTAG clones exhibited moderately reduced proliferation (Fig. EV1E), accompanied by an increase in migration and invasion, compared to the parental A673 cells (Fig. EV1F,G). The A2.2 dTAG clone exhibited colony-forming ability in soft agar comparable to that of parental cells, whereas B3.1 cells showed a reduced capacity for anchorage-independent growth (Fig. EV1H).

To titrate EF levels, we treated EF-dTAG clones with a dilution series of nanomolar dTAG$^V$-1 concentrations and assessed EF protein levels by Western blot, flow cytometry, and confocal microscopy (Figs. 1B,C and EV2A–C). We observed a gradual and highly reproducible loss of EF protein, decreasing by approximately 14% (1.5 nM), 35% (5 nM), 57% (15 nM), 84% (50 nM), and 93% (150 nM) at 24 h in both EF-dTAG clones (Figs. 1B and EV2A,C; Dataset EV3). Upon ligand washout, EF levels returned to baseline within 16 h (Fig. 1D). Together, these results validate A673 dTAG clones as a suitable model for studying EF fluctuations in EwS cells.

## Partial EF depletion significantly increases migration and invasion, with loss of colony formation in vitro

To investigate the impact of distinct EF levels on functional phenotypes in EwS, we pretreated EF-dTAG clones with increasing dTAG$^V$-1 concentrations for 24 h, followed by migration assays (Fig. 2A). A stepwise reduction of EF protein levels led to a gradual increase in the number of migratory EwS cells peaking at 15 nM dTAG$^V$-1 (≈43% EF, Dataset EV3). Further EF depletion began to reverse the effect on cell migration. Consistently, the invasive potential of EF-dTAG clones into collagen I was progressively enhanced upon gradual EF depletion, as assessed using an organ-on-a-chip platform. We observed a pronounced vessel-like sprouting phenotype, characterized by single, spindle-shaped invasive EwS cells leading subsequent invading cells (Fig. 2B). In contrast, cell proliferation and apoptosis rates remained unchanged by increasing dTAG$^V$-1 concentrations (Fig. EV3A,B).

Next, we evaluated the anchorage-independent growth capacity of dTAG$^V$-1-treated EF-dTAG cells in soft agar over a 3-week period as an in vitro surrogate of tumorigenic potential (Fig. 2C). Even a slight reduction in EF levels at 1.5 nM dTAG$^V$-1 (≈86% EF) was sufficient to decrease colony formation, while further EF depletion completely abolished it. Collectively, these results demonstrate that a moderate, but incomplete reduction of EF

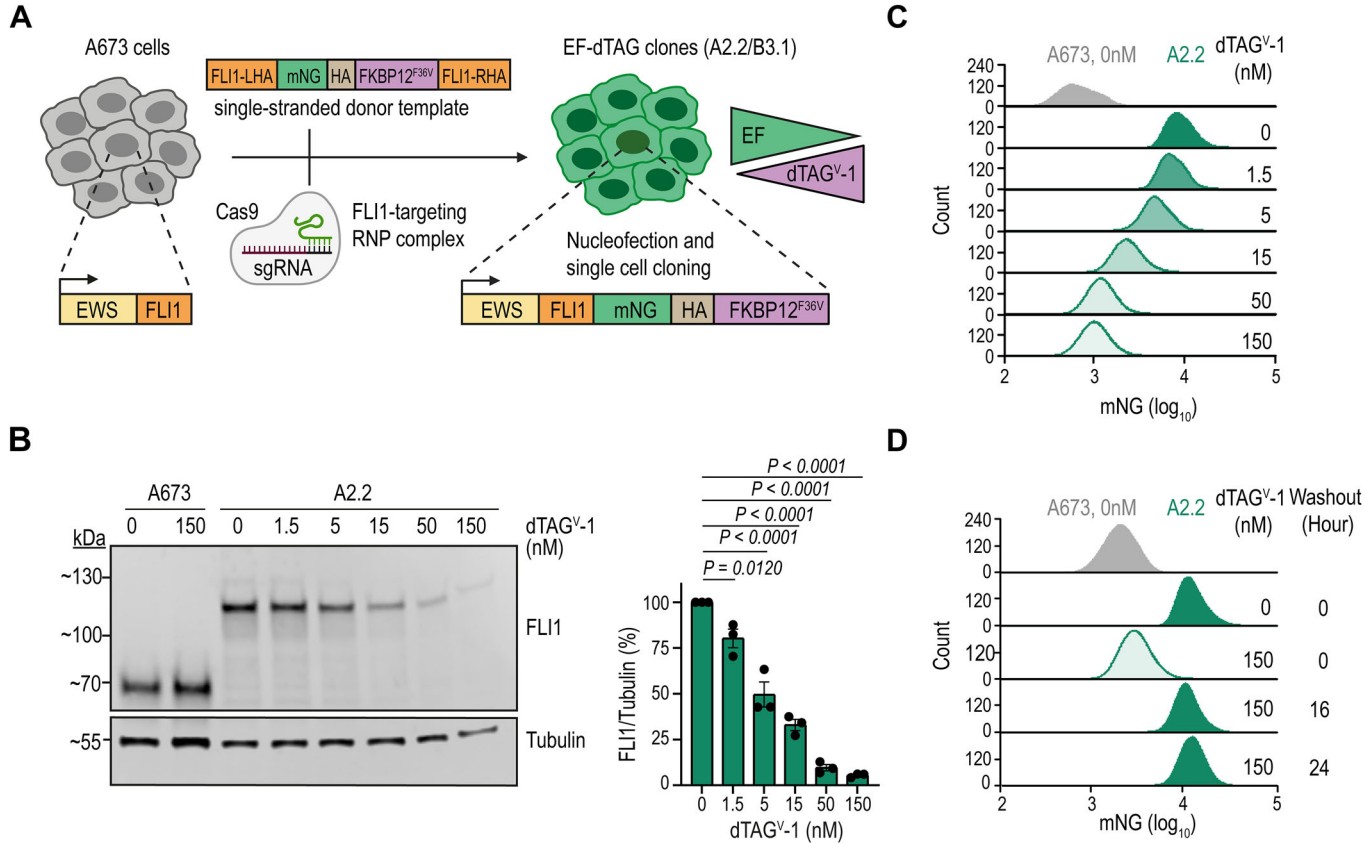

**Figure 1. Generation and establishment of EF thresholds.**

(A) Schematic representation of the A673 cell line editing strategy and EF titration using the dTAG system. sgRNA single-guide RNA, RNP ribonucleoprotein, mNG mNeonGreen. (B) Western blot analysis of FLI1 protein levels in EF-dTAG clone A2.2 and parental A673 cells treated with the indicated concentrations of dTAG$^V$-1 for 24 h (left). Tubulin was used as a loading control. Bar graphs depict EF protein levels normalized to Tubulin, presented as mean ± SEM (right; $n = 3$ independent biological replicates). $P$ values were determined using one-way ANOVA with post hoc Dunnett's multiple comparisons test. (C, D) Flow cytometry analysis of mNG fluorescence intensity in EF-dTAG clone A2.2 treated with increasing concentrations of dTAG$^V$-1 for 24 h (C) or following 16 or 24 h of washout after treatment with 150 nM dTAG$^V$-1 (D). Data shown are representative of at least three independent biological replicates. Source data are available online for this figure.

levels induces significant migratory and invasive characteristics in EwS cells, while anchorage-independent tumor cell growth is highly sensitive to minor decreases in EF thresholds.

## Gradual EF depletion reveals EF threshold-dependent gene sets

RNA-sequencing of EF-dTAG clones A2.2 and B3.1 after 24 h of continuous dTAG$^V$-1 treatment revealed a consistent transcriptomic response to increasing dTAG$^V$-1 concentrations, with 1,791 genes being differentially expressed at any dTAG$^V$-1 concentration in any of the two clones (DESeq2; 0 nM treatment as reference; FDR < 0.05 and absolute log$_2$ fold change >2; Fig. 3A; Dataset EV4). About 1109 genes showed a gradual increase in expression, whereas 682 genes decreased upon stepwise EF depletion (Fig. EV4A,B). In line with our phenotypic observations, gene set enrichment (GSE) analysis revealed that upregulated genes were associated with EMT, myogenesis, hypoxia and TNF-alpha signaling starting at 5 nM of dTAG$^V$-1 (Fig. 3B). Conversely, E2F targets and G2M checkpoint gene sets were enriched among the downregulated genes at intermediate to high dTAG$^V$-1 concentrations (≥15 nM).

To discriminate direct and indirect effects of graded EF depletion on de novo gene expression, we treated clone A2.2 with increasing dTAG$^V$-1 concentrations for only 3 h (Fig. EV4C) and performed thiol-(SH)-linked alkylation for the metabolic sequencing of RNA (SLAM-seq). This method enables the precise quantification of nascent RNA following dTAG$^V$-1 treatment by incorporating 4-thiouridine (4SU) into transcripts, resulting in thymidine-to-cytosine (T > C) conversions in RNA-sequencing reads (Herzog et al, 2017; Muhar et al, 2018) (Fig. EV4D; Dataset EV5). We found a total of 376 upregulated and 434 downregulated genes (Fig. 3C, DESeq2; FDR <0.05, absolute log$_2$ fold change >1). GSE analysis of the upregulated genes was consistent with the bulk RNA-seq data, revealing enrichment in pathways such as EMT, hypoxia and TNF-alpha signaling (Fig. 3D). In contrast, the downregulated genes did not result in any significant enrichment.

To identify genes with distinct immediate responses to stepwise EF depletion, we grouped DEGs into five gene clusters based on their response to increasing dTAG$^V$-1 ligand concentrations (Fig. 3E; Dataset EV6). Gene clusters G1 and G2 comprise EF-repressed targets that were upregulated upon dTAG$^V$-1 treatment. G1 genes

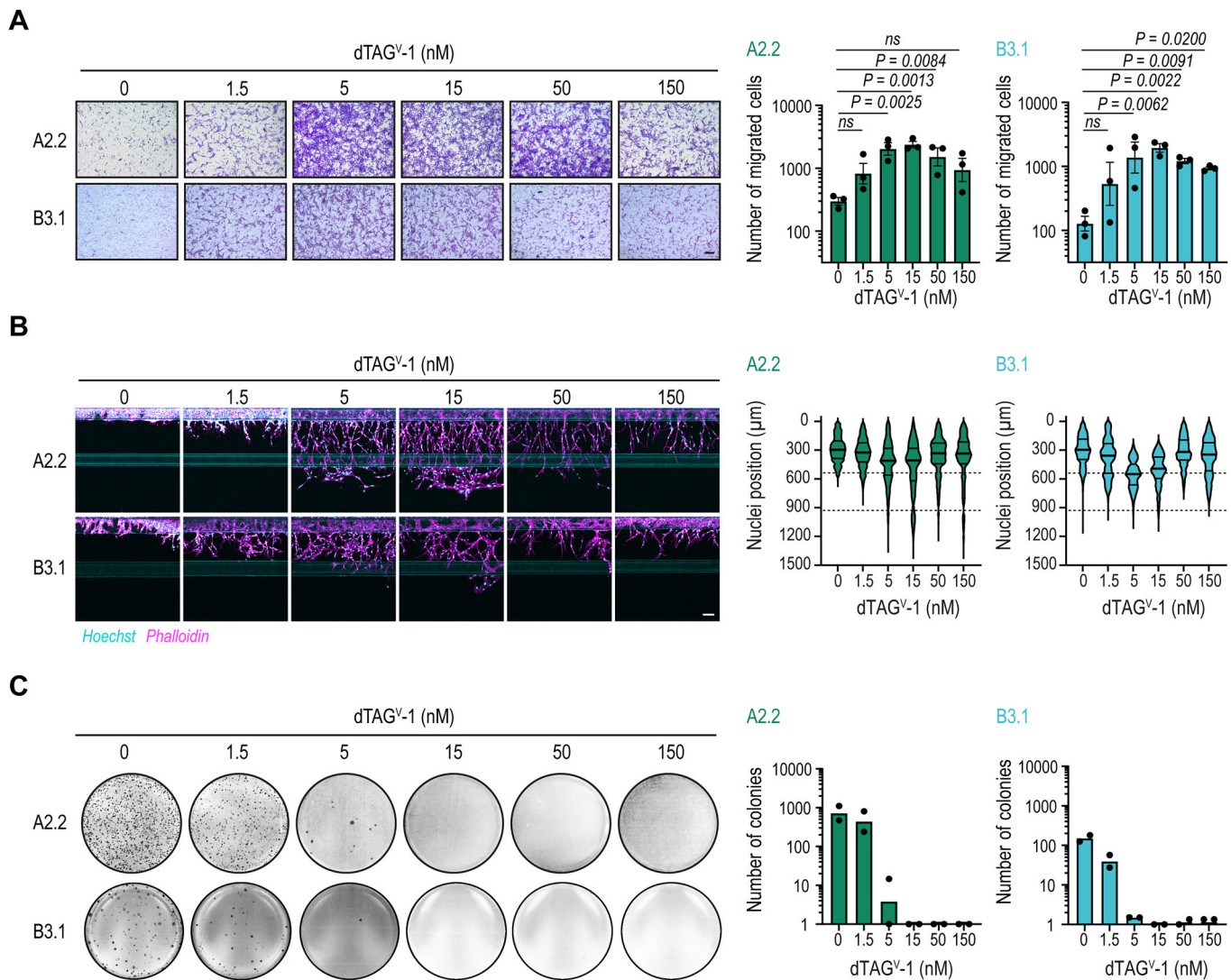

**Figure 2. Functional characterization of the EF gradient in EF-dTAG clones.**

(A) Transwell migration assay showing migrated EF-dTAG clones pretreated with dTAG$^V$-1 for 24 h at the indicated concentrations. Representative images from one of three independent biological replicates are shown (left). Migrated cell counts are presented as mean ± SEM after log10 transformation (right; $n = 3$). Scale bar = 200 μm. $P$ values were calculated using one-way ANOVA with Dunnett's post hoc multiple comparisons test. ns not significant. (B) Organoplate invasion assays depicting cell invasion into collagen I extracellular matrix (ECM) stained with Hoechst (cyan, nuclei) and phalloidin (magenta, actin cytoskeleton) 7 days after seeding and treating EF-dTAG clones at the indicated concentrations of dTAG$^V$-1. Representative images from one of two independent biological replicates are shown (left). Mean nuclei positions along the Y-axis are shown, with phase guides indicated by dotted lines at 535 and 930 μm (right; $n = 2$). Scale bar = 100 μm. (C) Soft agar colony formation assays showing crystal violet-stained colonies formed by A2.2 and B3.1 cells three weeks after seeding and treatment with dTAG$^V$-1 at the indicated concentrations. Representative images from one of two independent biological replicates are shown (left). Colony counts are presented as mean ($n = 2$) following log10 + 1 transformation (right). Source data are available online for this figure.

exhibited a gradual increase in expression from the lowest to the highest dTAG$^V$-1 concentration. In contrast, genes in cluster G2 exhibited a steep upregulation in response to dTAG$^V$-1 concentrations up to 15 nM, reaching a plateau at higher concentrations. These two EF-repressed clusters showed a strong enrichment in genes implicated in TNFα signaling, hypoxia, and EMT (Fig. EV4E). For EF-activated targets, we observed three patterns: genes in the smallest cluster G3 exhibited a steep decline in expression starting at the lowest dTAG$^V$-1 concentration (1.5 nM; ~14% EF reduction) continuing up to 15 nM (~57% EF reduction) (Fig. 3E). Interestingly, at higher dTAG$^V$-1 concentrations, where EF levels

approached near-complete depletion, a paradoxical upregulation in gene expression was observed. In contrast, genes in G4 and G5 decreased in expression upon gradual EF depletion. While the expression of G4 genes remained unchanged up to a dTAG$^V$-1 concentration of 5 nM ( ~ 35% EF reduction), G5 genes displayed a decline in expression already at 1.5 nM (~14% EF reduction). Notably, cluster G5 includes several well-characterized GGAA microsatellite-driven genes such as *BCL11B, CCND1, NKX2-2, NPY1R, NR0B1,* and *POU3F1* (Gangwal et al, 2008; Garcia-Aragoncillo et al, 2008). In summary, we have identified a set of 810 genes with distinct sensitivities to acute EF dosage change,

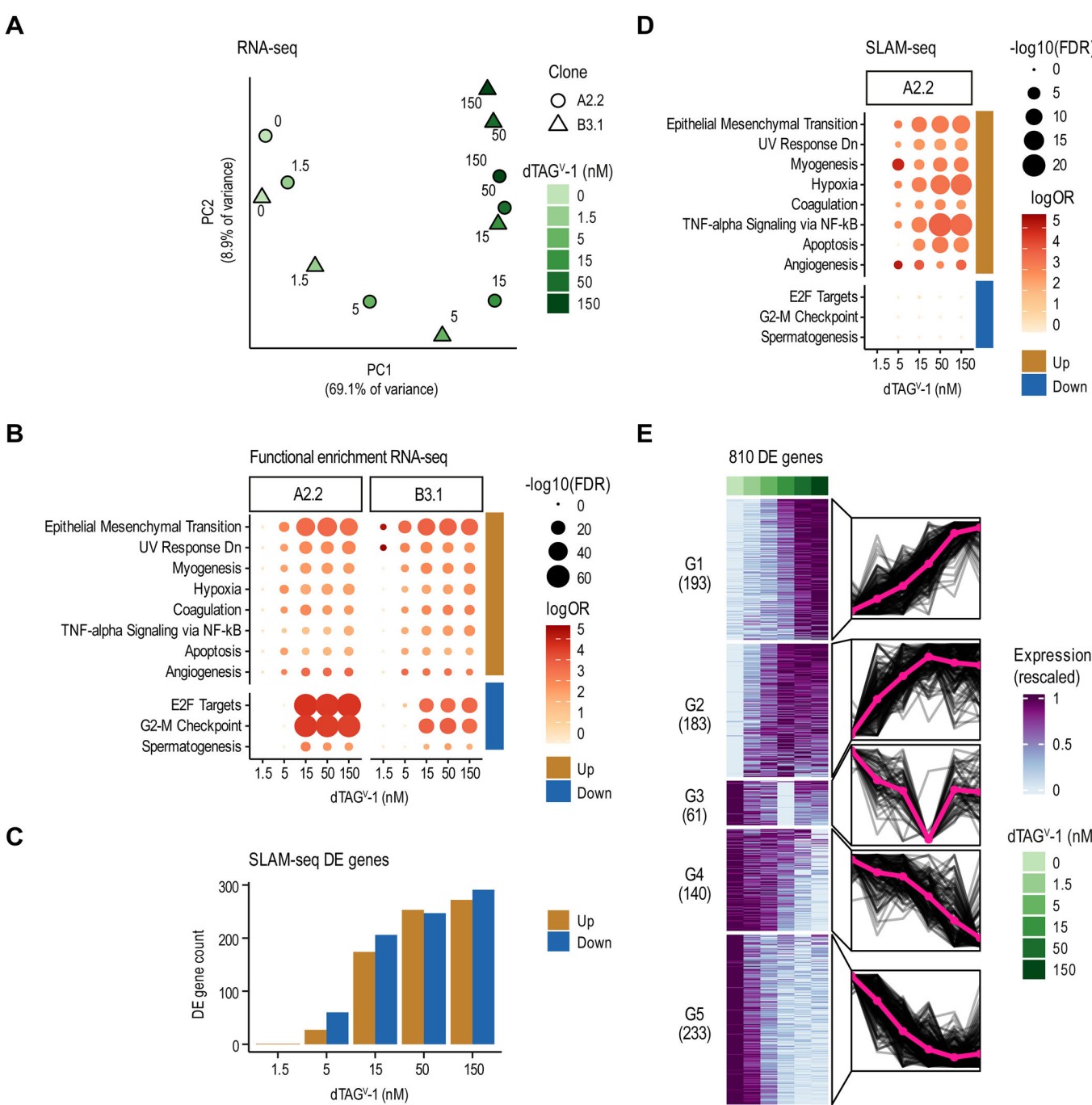

**Figure 3. Transcriptomic alterations induced by EF gradient.**

(A) EF level-dependent transcriptome changes analyzed by RNA-seq following 24-h dTAG^V-1 gradient treatment. Principal components of 1791 differentially expressed genes (DEGs) across both EF-dTAG clones and dTAG^V-1 concentrations (DESeq2); 0 nM treatment as reference; FDR <0.05 and absolute log₂ fold change >2). Each point indicates one condition after averaging technical replicates. Shape and color indicate the clonal cell line and the dTAG^V-1 concentration, respectively. (B) Functional enrichment of DEGs from panel (A), highlighting significantly enriched pathways from the MSigDB Hallmark 2020 database (hypergeometric test using hypeR; FDR <0.001, log₂ odds ratio >3; background: all DESeq2 tested genes). Circle size and color indicate the FDR-adjusted p value and the odds ratio, respectively. Genes were grouped by expression change direction as indicated by the colored bar on the ride side. (C) Bar plots showing the number of DEGs after 3 h of dTAG^V-1 treatment assessed by SLAM-seq (DESeq2; FDR <0.05, absolute log₂ fold change >1). Only nascent RNA reads with at least two T > C conversions were used for the analysis. (D) Functional enrichment of DEGs from panel (C). Analysis as in panel (B). (E) Identification of gene response groups by k-means clustering of expression profiles. Every gene from panel (C) was averaged across replicates, rescaled to range from 0 to 1 and plotted in a heatmap with treatment conditions on the x-axis and genes on the y-axis (left) and as individual lines grouped by cluster with conditions on the x-axis and rescaled expression on the y-axis (right). The bold pink line shows the cluster mean. The optimal number of clusters was determined using gap statistics. Source data are available online for this figure.

elucidating distinct EF dosage-dependent regulatory mechanisms and pathways.

## Distinct architecture of EF chromatin binding regions determines the pattern of response to EF modulation

The differences in the acute responses of individual gene expression clusters to gradual EF depletion may result from variable EF binding affinities to genomic regions with distinct chromatin architectures and from functional interactions with other transcription cofactors. We therefore conducted genome-wide profiling of EF-bound chromatin regions and associated histone landscapes in parental A673 cells and EF-dTAG clone A2.2 using CUT&RUN with antibodies directed to the FLI1 C-terminus, H3k27ac, H3k27me3, and H3k4me3. We identified a total of 12,370 consensus EF binding peaks, supported by replicates in either parental A673 or A2.2 cells (Fig. EV5A; Dataset EV1). The majority of EF binding was observed in promoters (44.5%) characterized by H3K27ac and H3K4me3 chromatin marks, followed by intronic (28.5%) and distal intergenic (20.4%) regions marked by H3K27ac enhancer marks (Fig. EV5B,C). In line with prior findings, FLI1-binding peaks showed significant depletion in regions marked by the repressive H3K27me3 mark (Fig. EV5C) (Tomazou et al, 2015).

We scanned the EF-bound chromatin regions for occurrences of transcription factor binding motifs from the JASPAR2020 core collection (Fornes et al, 2020) and identified eight peak clusters based on recurrent motif associations (Fig. 4A–C; Dataset EV7). The largest cluster, P1, contained 4160 peaks that were predominantly located at promoter regions and enriched with binding motifs for Krüppel-like and other zinc finger transcription factors (KLF15/16, SP9, MAZ, and ZNF148). Peak cluster P3 ($n = 2079$) showed significant enrichment for canonical monomeric ETS binding motifs. A distinct cluster, P5 (958 peaks), was uniquely characterized by a high enrichment of EF binding GGAA repeat motif matches. These peaks were broader, exhibited stronger signal intensity than those in other clusters and were predominantly located in intergenic regions. In addition, they were most sensitive to low dTAG$^V$-1 concentrations as measured by fold change (Fig. EV5D).

To link EF binding regions with immediate response genes from our SLAM-seq analysis, we mapped the EF-binding peaks to the closest transcription start sites (≤150 kb upstream or downstream) of actively expressed genes and tested for association between gene clusters and peak clusters (Fig. 4D,E). We found that EF-repressed genes in clusters G1 and G2 were significantly depleted near P5 peaks, i.e., peaks with intergenic GGAA microsatellites (Fig. 4E). Instead, genes within cluster G1 were associated with the promoter- and KLF-linked peak cluster P1. Interestingly, EF-repressed genes with the highest sensitivity to EF depletion (G2) were enriched near ETS-linked peak cluster P3. In contrast, EF-activated genes in G5 were significantly enriched near GGAA microsatellite-linked peaks in cluster P5, and depleted near peak clusters P1 and P3, suggesting a distinct gene regulatory mechanism compared to EF-repressed genes.

## Sustained activation of EMT-associated transcriptional programs upon transient EF modulation

After establishing the acute response of EwS cells to gradual EF depletion, we investigated the molecular and phenotypic consequences of prolonged EF dosage modulation. A2.2 cells were treated with the dTAG$^V$-1 gradient for 7, 14, 21, and 28 days, and B3.1 cells were treated for 21 days. To mimic dynamic EF fluctuations, a subset of cells underwent a 7-day washout of the ligand following the treatment period. For each condition, samples were collected for RNA-sequencing. Flow cytometry analysis demonstrated that the EF gradient remained stable across 7, 14, 21, and 28 days of dTAG$^V$-1 treatment in EF-dTAG A2.2 clone (Fig. EV6A). Upon washout, baseline EF expression levels were restored within 16 h (Fig. 1D) and remained stable at the end of the 7-day washout period (Fig. EV6B). Principal component analysis of the RNA-seq data demonstrated a clear separation between dTAG$^V$-1-treated samples and untreated controls (Figs. 5A and EV6C), with gene expression changes correlating with both dTAG$^V$-1 concentration and treatment duration (Fig. 5B; Dataset EV8). GSE analysis revealed EMT to be the top-significant pathway irrespective of the dTAG$^V$-1 concentration or treatment durations in both the EF-dTAG clones (Figs. 5C and EV6D). Washout samples were clearly separated from treated samples, but the complete recovery of EF levels only partially reversed the observed gene expression changes. The largest number of dysregulated DEGs were seen after prolonged exposure to higher dTAG$^V$-1 concentrations (Figs. 5D and EV6E left; Dataset EV8). Across all treatment durations and concentrations, most of the sustained effects could be attributed to EF-repressed genes, which remained upregulated after rescue of fusion protein expression (Figs. 5D and EV6E left). EMT, TNF-alpha signaling, hypoxia, and myogenesis-related gene sets were significantly enriched, displaying a consistent response pattern that remained largely independent of dTAG$^V$-1 concentration in cells treated for 21 or 28 days, followed by a 7-day EF rescue in both EF-dTAG clones (Figs. 5E and EV6E right). A total of 106 genes were consistently upregulated in both EF-dTAG clones following EF rescue after 21-day treatment at any dTAG$^V$-1 concentration. Notably, 23 of these genes were associated with the EMT gene set from the MSigDB Hallmark 2020 pathways (Fig. 5F). These results demonstrate that EF modulation persistently altered the EwS transcriptome, fostering an EMT transcriptional program.

## EwS cells retain increased metastatic potential upon transient EF modulation

To validate the transcriptomic findings at a functional level, both EF-dTAG clones were treated with gradient concentrations of dTAG$^V$-1 over 7 or 21 days, followed by a 7-day ligand withdrawal. Colony formation, cell migration, and invasion assays were then performed. The washout of dTAG$^V$-1, after a 7-day pretreatment with the ligand, fully rescued the EF gradient-dependent migration phenotype, restoring it to the level observed in untreated cells (Fig. EV7A). However, following a 21-day pretreatment with dTAG$^V$-1 ligand, withdrawal for 7 days was insufficient to fully reverse the migration phenotype. This incomplete recovery was observed at dTAG$^V$-1 concentrations of 5–150 nM for clone A2.2 and 5–15 nM for clone B3.1 (Figs. 6A and EV7B). Prolonging the ligand washout period to two weeks also failed to fully revert the migratory potential of 21-day dTAG$^V$-1-treated EF-dTAG clones to that of untreated controls (Fig. EV7C,D). Likewise, organ-on-a-chip invasion assays demonstrated a substantial retention of invasive properties of 21 days dTAG$^V$-1 treated EF-dTAG clones after 7 days

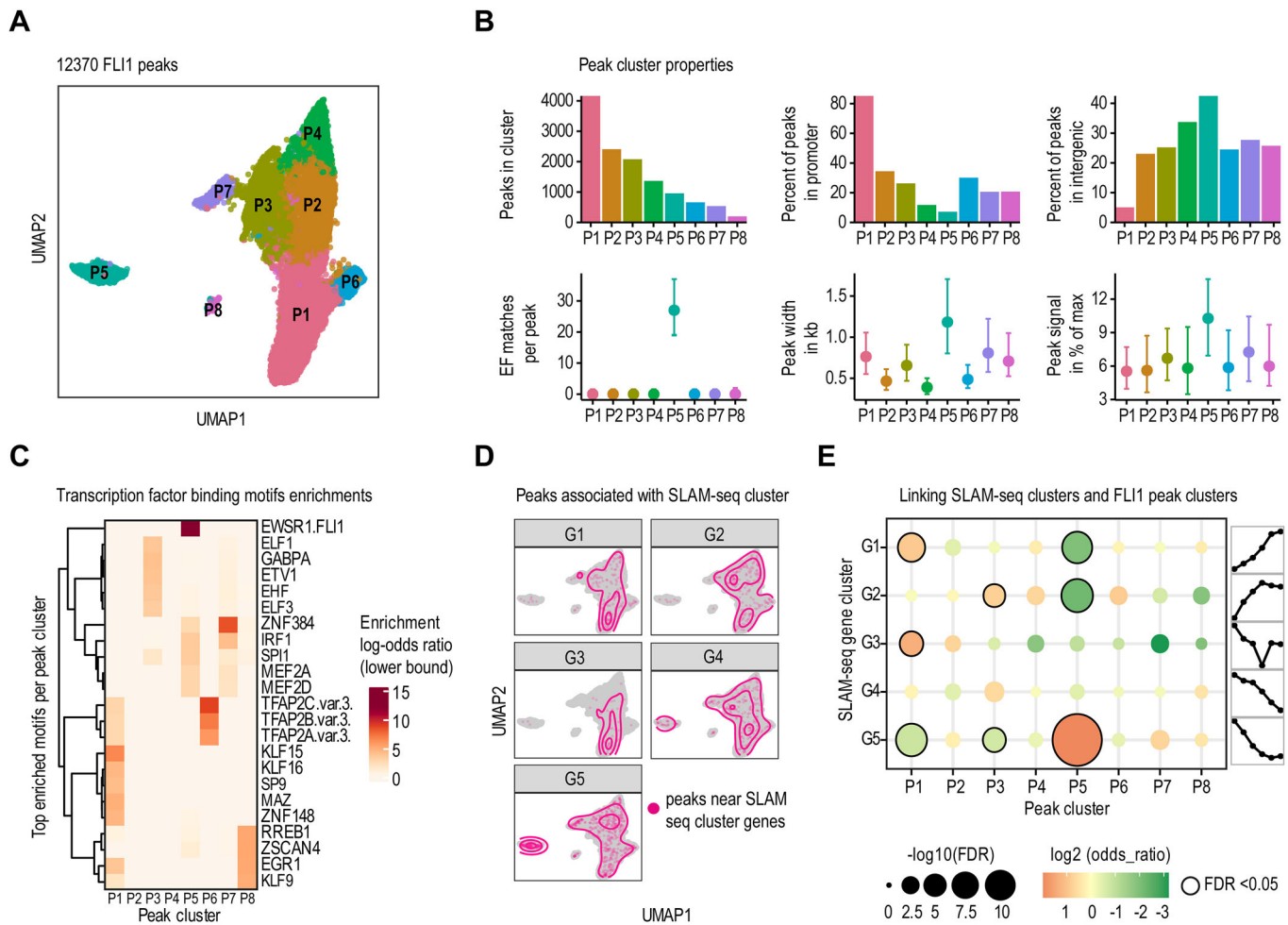

Figure 4. Profile of EF-bound chromatin regions and its association with immediate response genes.

(A) UMAP plots summarizing the results of a genome-wide analysis of EF-bound chromatin regions with CUT&RUN. Each point indicates one peak. A total of 12,370 consensus EF binding peaks were identified, supported by replicates in either parental A673 or A2.2 cells. JASPAR2020 (Fornes et al, 2020) human core transcription factor binding motif matches were counted for every peak, and the Manhattan metric was used to define peak-to-peak distances. Clustering was done with the Louvain method (Blondel et al, 2008) and eight distinct EF-binding peak clusters were identified. (B) For each peak cluster, the total number of peaks (P1 = 4160, P2 = 2409, P3 = 2079, P4 = 1367, P5 = 958, P6 = 661, P7 = 538, and P8 = 198), their overlap with promoters or intergenic regions, number of matches to the EF binding motif (matrix profile MA0149.1), width, and CUT&RUN signal intensity (MACS2 reported peak signal rescaled to [0, 100] per sample, replicates averaged were summarized (median and interquartile range shown as points and ranges). (C) Transcription factor binding motif match enrichments were calculated for every peak cluster, displaying the top five motifs. Motifs were filtered (one-sided Fisher's exact test; FDR <0.01, lower bound of 95% confidence interval of log2 odds ratio >2) and ranked by the log2 odds ratio. (D) Density contour plots overlaid on the UMAP plot from panel (A), indicating association of peaks with genes responsive to EF degradation. Every peak was linked to its closest response gene (out of the 810 differentially expressed SLAM-seq genes, cp. Fig. 3E) within 150 kb distance. (E) Cluster-specific enrichments were calculated for each EF-dependent gene cluster (G1–G5) and each EF-binding peak cluster (P1–P8) using two-sided Fisher's exact test. Circle size and color indicate the FDR and odds ratio, respectively. Significant associations (FDR <0.05) are highlighted with a solid border. Source data are available online for this figure.

of EF recovery in the organoplate (Figs. 6B and EV7E). While dTAG$^V$-1-treated cells predominantly exhibited a dispersed sprouting mode of invasion, the washout cells displayed a markedly different pattern characterized by collective, clustered invasion fronts. Consistent with the consequences of short-term dTAG$^V$-1 treatment, soft agar colony formation of EF-dTAG clones decreased with gradual EF modulation for 21 days and was completely abolished at dTAG$^V$-1 concentrations above 5 nM (Figs. 6C and EV7F). However, when these 3-week pretreated cells were seeded and subsequently maintained in the absence of dTAG$^V$-1 throughout the soft agar period, an evident recovery of colony-forming ability was observed (Figs. 6C and EV7F). These findings

indicate that transient EF modulation can condition cells to maintain migratory and invasive phenotypes, while preserving clonogenic growth.

We hypothesized that these phenotypic changes in cells were linked to an increased capacity to metastasize in vivo. To investigate this hypothesis, we first examined the extravasation properties of A2.2 cells following partial EF depletion and subsequent rescue using a zebrafish model. A2.2 cells were treated with 5 nM or 15 nM dTAG$^V$-1 for 3 weeks before being intravascularly injected into zebrafish larvae with no further dTAG$^V$-1 ligand provided, thus allowing recovery of the fusion protein. Additionally, we included an EF rescue condition prior to

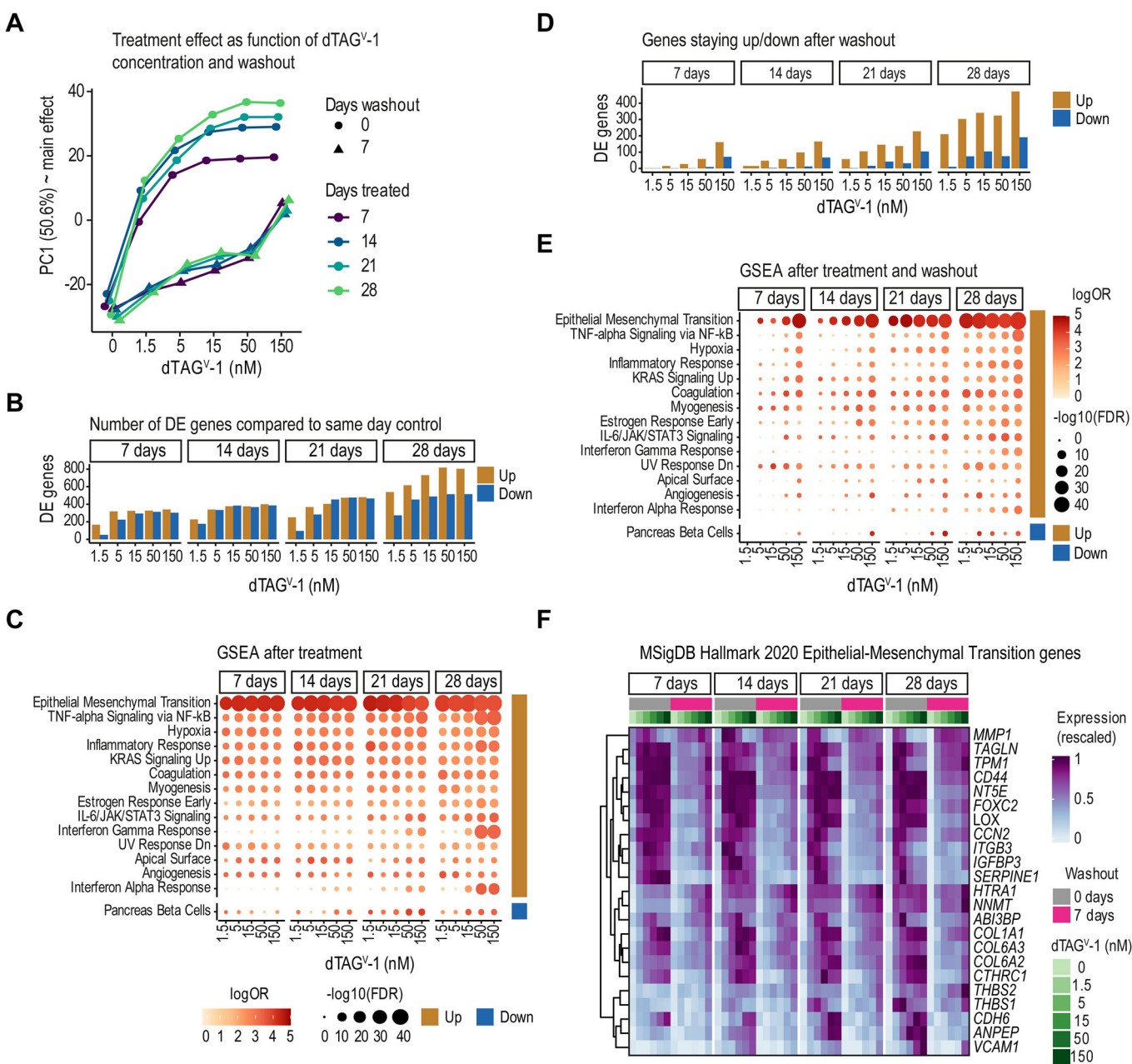

**Figure 5. Transcriptional effects of prolonged and transient EF modulation.**

(A) Principal component analysis (PCA) of 1455 highly variable genes (gene expression with standard deviation >0.5 after normalization with DESeq2 vst function and averaging two replicates), highlighting the primary transcriptional effects of prolonged and transient EF dosage modulation (RNA-seq). The plot shows the value of principal component 1 (PC1), the main treatment effect, at different dTAG$^V$-1 concentrations. Point shape distinguishes conditions without (circle) and with 7-day washout (triangle), and color indicates the number of days of dTAG$^V$-1 treatment. (B) Number of DEGs following dTAG$^V$-1 gradient treatment compared to untreated controls on the same day (DESeq2; FDR <0.05, absolute log$_2$ fold change >2). (C) Functional enrichment analysis of DEGs after treatment, showing MSigDB Hallmark 2020 pathways that are significantly enriched in any of the dTAG$^V$-1 treatments (hypergeometric test; FDR <0.001, log$_2$ odds ratio >3; background: all DESeq2 tested genes). Circle size and color indicate the FDR-adjusted $p$ value and the odds ratio, respectively. (D) Number of DEGs following treatment and subsequent washout compared to untreated control on the same day (DESeq2; FDR <0.05, absolute log$_2$ fold change >2). (E) Functional enrichment of DEGs after dTAG$^V$-1 treatment followed by washout (cf. C). (F) Heatmaps depicting genes from the MSigDB Hallmark 2020 epithelial-mesenchymal transition pathway that are upregulated after a 3-week treatment and persist following ligand washout for at least one dTAG$^V$-1 concentration in clone A2.2 and B3.1. Expression values are averages of replicates after normalization (cf. A). Conditions on the x-axis are ordered by treatment duration, washout status, and dTAG$^V$-1 concentration, as indicated above the heatmaps. Source data are available online for this figure.

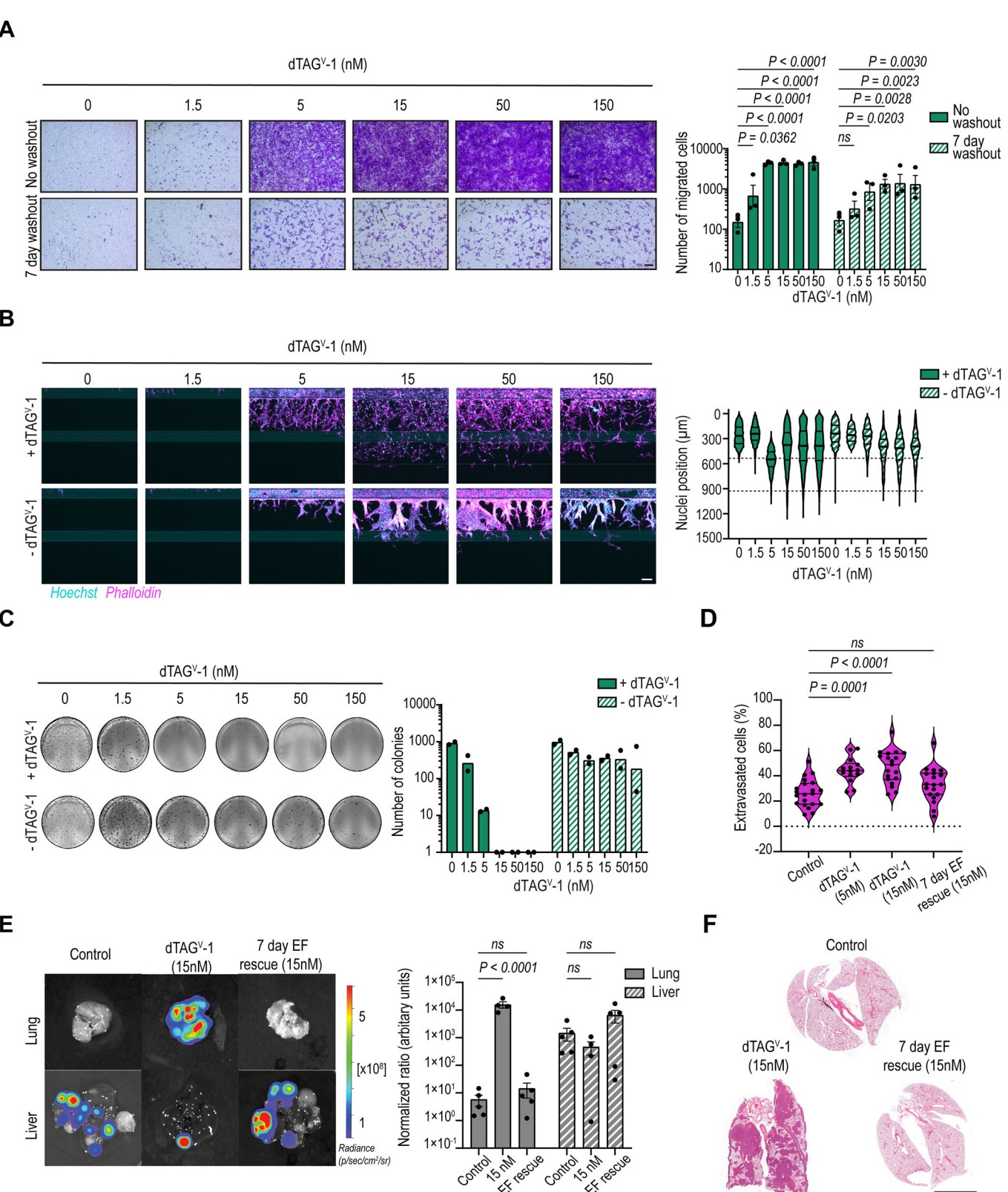

**Figure 6.   In vitro and in vivo assessment of metastatic effects following transient EF modulation.**

(A) Transwell migration assays showing migrated A2.2 cells pretreated with dTAG$^V$-1 for 21 days at the indicated concentrations or pretreated for 21 days followed by a 7-day washout. Representative images from one of three independent biological replicates are shown (left). Migrated cell counts are presented as mean ± SEM after log10 transformation (right; $n = 3$). Scale bar = 200 μm. (B) Organoplate invasion assays depicting cell invasion into collagen I ECM 7 days after seeding A2.2 cells treated with dTAG$^V$-1 for 21 days at the indicated concentrations, with or without further dTAG$^V$-1 treatment. Representative images from one of two independent biological replicates are shown (left). Mean nuclei positions along the Y-axis are shown, with phase guides indicated by dotted lines at 535 and 930 μm (right; $n = 2$). Scale bar = 100 μm. (C) Soft agar assays showing colonies formed by A2.2 cells three weeks after seeding 21 days dTAG$^V$-1-treated cells at indicated concentrations with or without further dTAG$^V$-1 treatment. Representative images from one of two independent biological replicates are shown (left). Colony counts are presented as mean ($n = 2$) following log10 + 1 transformation. (D) Zebrafish extravasation assays 3 days post injection of cells treated with dTAG$^V$-1 for 21 days at the indicated concentrations, or 21 days of 15 nM dTAG$^V$-1 followed by a 7-day washout. Extravasation data were presented as a percentage of extravasated cells ($n = 21$ for control, $n = 14$ for 5 nM, $n = 19$ for 15 nM, and $n = 19$ for 15 nM with EF rescue; $n$ = technical replicates). (E) Endpoint In vivo Imaging System (IVIS) imaging of lungs and livers from mice tail vein injected with A2.2 cells treated with 15 nM dTAG$^V$-1 for 21 days, with or without a 7-day washout, with no further dTAG$^V$-1 treatment. Representative images from one of the technical replicates are shown (left). Quantification of organ-wide luciferase intensity from tumors in the lungs and livers, normalized to whole-body luminescence of their respective conditions at day 0 (right, $n = 5$ mice/group, $n = 4$ mice for 15 nM). (F) H&E-stained lung sections from the conditions shown in panel E. Representative images from one of the technical replicates are shown. Scale bar = 5000 μm. P values for panels (A, E) were calculated using two-way ANOVA with Dunnett's post hoc multiple comparisons test. P values for panel (D) were calculated using one-way ANOVA with Dunnett's post hoc multiple comparisons test. ns not significant. Source data are available online for this figure.

injection by culturing the 15 nM dTAG$^V$-1 pretreated cells without ligand for 7 days prior to injection. At 3 days post injection, a significant increase in the number of extravasated tumor cells was observed in both the 5 nM and 15 nM dTAG$^V$-1 pretreated groups compared to untreated controls (Figs. 6D and EV7G). In contrast, the 7-day EF rescue condition did not exhibit significant differences in extravasation levels compared to untreated controls.

Next, we injected luciferase-expressing A2.2 cells with or without 21-day pretreatment (15 nM dTAG$^V$-1) into the tail vein of NSG mice. Again, we included an EF rescue condition where dTAG$^V$-1 treatment was stopped 7 days prior to injection (Fig. EV7H). Animals were subsequently monitored for 32 days in the absence of dTAG$^V$-1 to allow in vivo recovery of the EF fusion protein. Independent of treatment, all mice injected with EF-dTAG cells showed a comparable tumor burden in the liver, but cells pretreated with 15 nM dTAG$^V$-1 exhibited a significantly increased lung metastatic burden compared to untreated controls (Fig. 6E). These findings align with observations from the zebrafish extravasation model and suggest that transient EF protein modulation promotes tumor cell seeding and overt colonization of lung tissue in mammals. Histological analysis of H&E-stained lung sections confirmed enhanced metastatic burden in mice injected with dTAG$^V$-1 pretreated cells (Fig. 6F). Interestingly, despite retaining an elevated EMT transcriptional signature and increased in vitro migratory and invasive potential, the 7-day EF recovery condition did not significantly result in increased lung metastasis following tail vein injection (Fig. 6E). Together, these results suggest that after transient EF modulation, 7 days EF rescue prior to in vivo experimentation eliminates increased extravasation properties of EwS cells and their ability to develop lung metastases, while their invasive and migratory propensity remains increased.

## Discussion

In this study, we investigated the direct relationship of oncoprotein dosage and cellular plasticity by the example of EwS at both molecular and functional levels. Knock-in of a fluorescent and a degron tag at the FLI1 carboxy-terminus was well tolerated in the primary non-metastatic tumor-derived A673 cell line, with minimal transcriptomic variation observed when compared to

single cell clones from unedited parental A673 cells. Although the EF carboxy-terminus was previously shown to play a role in protein interactions involved in cooperative DNA binding with other transcription factors (Kim et al, 2006), we did not observe major differences in genome-wide EF chromatin occupancy between the tagged and the untagged oncoprotein in A673 cells. However, our unsuccessful attempts to establish a similar model in other EwS cell lines suggest that tolerance to carboxy-terminal EF modification may depend on the underlying genomic or epigenomic background, reflecting a cellular adaptive response to maintain EF function, likely shaped by each line's unique (epi-)genetic context. The precise molecular mechanisms for this selectivity remain to be determined. Therefore, the restriction to a single cell line has to be seen as the major limitation of our study.

A key advantage of our degron model is its stability and reproducibility in revealing distinct phenotypic states associated with precisely defined and sustained EF levels. Using this model, we demonstrate for the first time that already a minor reduction in EF levels by as little as 14% was sufficient to induce a migratory and invasive phenotype in EF-dTAG cells, which peaked at an approximately 57% EF reduction. However, upon further EF depletion (84-93%) migratory and invasive capabilities markedly decreased, suggesting that a critical threshold of EF is required for sustaining pro-metastatic potential. Remarkably, even minimal EF modulation (around 14% EF reduction) decreased soft agar colony-forming ability, while no effects on cell proliferation were observed. Our study identifies precise EF thresholds that govern EMT-like transitions in A673, refining insights previously suggested by crude EF knockdown experiments (Chaturvedi et al, 2012; Franzetti et al, 2017; Katschnig et al, 2017). Furthermore, we extend and refine the "Goldilocks principle" of EF dosage-dependent regulation initially proposed for EwS cell growth (Seong et al, 2021) to include pro-metastatic tumor cell properties. Of note, despite the restriction of our model to A673 cells, the EF-linked phenotypic changes in cell adherence and migration have previously been reported in SK-N-MC (Franzetti et al, 2017) and validated in TC71 and EWS502 EwS cells (Chaturvedi et al, 2012). Additionally, using independent perturbations, the role of EF in driving EMT, dormancy and stemness has been studied in EW-8, ES6 and ES8 cells (Khoogar et al, 2022), supporting the general relevance of EF-driven plasticity across EwS contexts.

To understand the molecular basis of EF level-dependent phenotypic plasticity, we monitored the acute and direct transcriptional response to graded EF depletion, hypothesizing that EF affinity to distinct target gene sets may vary in a concentration-dependent manner. Indeed, we found clusters of genes with distinct response patterns to graded EF depletion. We found GGAA microsatellite-driven EF-activated genes (G5) to be the most sensitive to even subtle (~14%) EF modulation, with near-complete transcriptional loss observed at 57% EF protein reduction. This aligns with the unique binding mechanism of EF at these sites, which engages GGAA repeat arrays via multimerization and biomolecular condensation (Chong et al, 2018; Zuo et al, 2021). Given that endogenous nuclear EF concentrations in EwS cells are estimated at ~200 nM (Chong et al, 2018), prior studies demonstrating impaired in vitro condensate formation when lowering EF protein concentrations from 250 to 100 nM (Zuo et al, 2021) support our ex vivo observation that even modest reductions in EF levels disrupt GGAA microsatellite-associated target gene activation. In sharp contrast, EF-activated gene cluster G4 was much more robust and poorly sensitive to acute EF loss of up to 35%, followed by an EF concentration-dependent gradual decrease of gene expression, with complete loss observed only upon near-complete EF depletion. Lack of enriched transcription factor motifs at corresponding EF-bound regions may suggest involvement of other unidentified EF cofactors or an early indirect mechanism of gene regulation.

Among the EF-repressed genes, those characterized by the presence of monomeric canonical ETS binding sites near EF-bound chromatin (G2) showed the highest sensitivity to subtle EF modulation, with maximum derepression occurring at ~57% EF reduction. Of note, these genes completely lacked association with EF-bound GGAA microsatellites. This pattern is consistent with previous reports suggesting two distinct binding site preferences of EF in EwS, where EF-activated sites typically contain four or more consecutive GGAA repeats, whereas EF-repressed sites are often linked to non-repetitive, canonical ETS motifs (Guillon et al, 2009; Johnson et al, 2017; Riggi et al, 2014). However, unlike studies that mainly associate gene activation with EF binding at GGAA microsatellites within promoter regions (Gangwal et al, 2008; Johnson et al, 2017), our data indicate a predominant association with intergenic regions consistent with findings that GGAA microsatellites act as potent distal enhancers in EwS (Boulay et al, 2018; Guillon et al, 2009).

The lack of enrichment for canonical ETS and E2F binding motifs among EF-activated DEGs in SLAM-seq experiments came as a surprise, as previous studies on EF chromatin binding have shown cooperative binding of EF with E2F3 in the promoters of E2F-regulated cell cycle genes (Bilke et al, 2013). Knockdown of EF has been shown to result in the replacement of activating E2F3 by the repressive E2F4 family member (Schwentner et al, 2015). Although we observed reduced EF binding to canonical ETS binding motifs already at 3 h of dTAG$^V$-1 treatment (EF binding peak cluster P3, Fig. 4E), we speculate that E2F3 replacement may require more time to affect the expression of cell cycle genes. Indeed, after 24 h of EF modulation, E2F target and G2/M checkpoint genes were most significantly enriched at EF reductions of at least 57%, suggesting robustness of EF-driven E2F target gene regulation to subtle short-term variations of EF expression.

EF-repressed immediate response genes lacking both canonical ETS and GGAA microsatellite motifs represented the second largest gene expression cluster (G1) and showed a graded response to stepwise EF depletion. A significant enrichment of binding motifs for Krüppel-like three-zinc finger transcription factors was found predominantly in promoters of G1 cluster genes. Among them was the transcriptional repressor KLF15, which has previously been implicated in the core regulatory circuitry of EF-interacting transcription factors, activated through EF binding to a GGAA microsatellite super-enhancer (Shi et al, 2020) and expressed as part of gene expression cluster G5. Its potential role in EF-mediated transcriptional repression remains to be established. Additionally, binding motifs for zinc finger transcription factors MAZ and ZNF148 were found enriched in promoter-associated EF binding peaks of G1 cluster genes. So far, nothing is known about functional interactions between EF and any of these candidate EF-interacting transcription factors and experimental validation is warranted in follow-up studies.

A particularly intriguing group of acute EF response genes, which also include *MYC* and its target genes (i.e., *TFB2M*, *HNRNPC,* and *SNRPD1*), exhibits a bimodal expression pattern that is dependent on EF thresholds (G3). These genes display a sharp decline in expression following subtle EF reductions, similar to the behavior observed in the GGAA microsatellite-driven cluster G5, with complete loss of expression occurring at residual EF levels of about 43%. However, upon further EF depletion, the expression of these genes resurges. A potential explanation for this increase in gene expression at very low EF concentrations may be reactivation by an EF-suppressed transcription factor, which remains to be defined. EF binding peaks in the vicinity of these genes were enriched in the same transcription factor binding motifs (P1) as G1 cluster genes and did not show a particular preference for either promoter or intergenic regions. Interestingly, the bimodal expression response of G3 cluster genes to EF depletion inversely correlated with variations in migratory and invasive phenotypes of EF-dTAG cells. These phenotypes peaked at an EF threshold of 43% and declined as G3 cluster gene expression increased at higher dTAG$^V$-1 concentrations. Although GSE analysis did not reveal specific functional annotations for this cluster, the G3 genes may represent critical regulators of metastasis, warranting further investigation in future studies. While our study in A673 cells provides proof-of-principle that EF dosage modulates discrete transcriptional programs and migratory/invasive phenotypes, there may be inter-individual variation in EF-dose sensitivity and fluctuations across genetic backgrounds.

Our findings indicate that EwS tumor phenotype is not solely dependent on the binary presence or absence of EF but is instead influenced by EF dosage levels and fluctuations thereof as previously suggested (Aynaud et al, 2020; Franzetti et al, 2017; Khoogar et al, 2022; Wrenn et al, 2025). The pace at which EF fluctuates naturally or in response to EF-modulatory treatment remains unknown. We leveraged the capability of the dTAG degron system to rapidly re-establish EF expression after dTAG$^V$-1 ligand withdrawal, following transient dTAG$^V$-1-mediated graded EF degradation. This system allowed us to simulate EF fluctuation that can occur stochastically or during an EF-directed therapeutic approach and examine the dynamics of EF target gene recovery and its impact on tumorigenesis. Notably, transient EF degradation for

1–4 weeks, followed by its re-expression for 1 week, led to a sustained transcriptional dysregulation of mostly EF-repressed EMT-related genes, irrespective of EF levels. Importantly, these persistent changes did not involve the entire EF program but instead a limited subset of genes, consistent with a state-dependent "memory" rather than global irreversibility. Consistent with prior studies showing rapid re-equilibration of EF heterogeneity at the population level (Franzetti et al, 2017; Wrenn et al, 2025), our model recapitulated this dynamic behavior following flow-sorting of EF-mNG^Dim and EF-mNG^High cells. As shown in Appendix Fig. S2A–C, both sorted populations progressively regenerated heterogeneity, with EF-mNG^Dim cells re-equilibrating after approximately 112 h and EF-mNG^High cells after 72 h. Therefore, our findings refine earlier models and suggest that both the duration and magnitude of EF fluctuation determine whether cells revert or transition into a metastable invasive state. Among the dysregulated EMT-associated genes, we identified several previously validated direct EF targets, including *CD44* (Fernández-Tabanera et al, 2023), *TAGLN* (Katschnig et al, 2017), and *PHLDA1* (Boro et al, 2012). In addition, our SLAM-seq data revealed novel candidate targets such as *SEMA3C*, *COL1A1*, *HRH1*, *FILIP1L*, *FOSL2*, and *S100A10*, expanding the repertoire of EF targets potentially involved in EwS EMP (Kwon et al, 2016; Li et al, 2021; Park et al, 2023; Wang et al, 2023b; Yin et al, 2019; Zhang et al, 2018). Additionally, we find clinically actionable candidates such as *NNMT*, *CDCP1*, and *ANPEP*, which are yet to be functionally validated in other EwS cell lines, among persistently upregulated genes in A673 EF-dTAG cells (Jenkins et al, 2011; Neelakantan et al, 2018; Wong et al, 2023). In vitro assays revealed a significant retention of an elevated migratory and invasive phenotype and partial retention of colony-forming ability in soft agar following 7-day EF rescue after three weeks of EF modulation. Notably, in organ-on-a-chip invasion assays, cells under sustained EF suppression exhibited features of a complete EMT phenotype, characterized by single spindle-shaped cells acting as invasion path-generating pioneers. In contrast, following dTAG^V-1 ligand washout and restoration of full EF expression, cells predominantly adopted a hybrid epithelial/mesenchymal (E/M)-like phenotype, invading collectively as cohesive clusters. This phenotypic switch is of potential clinical relevance, as hybrid E/M states have been implicated in the generation of circulating tumor cell (CTC) clusters in the blood that exhibit resistance to apoptosis and possess up to 50-fold greater metastatic potential compared to individually disseminated CTCs undergoing complete EMT (Aceto et al, 2014). Silveira et al previously proposed EF as a "phenotypic stabilizing factor" where ZEB2 and Claudin-1 plays a critical role in these hybrid E/M cell states (Silveira et al, 2023). Also, a recent report demonstrates that autocrine TGFβ2 signaling reinforces a stable hybrid CAF-like state in EF-high cells via derepression of *TGFBR2* and fusion-mediated antagonism of gene repression (Wrenn et al, 2025). Supporting these findings, our bulk RNA-seq data revealed derepression of several EMP-related genes, including epithelial markers such as Claudin-1 and Occludin, upon EF restoration after 21–28 days of treatment with dTAG^V-1 concentrations higher than 5 nM, while mesenchymal markers such as CD44 and CD73 remained elevated. These findings suggest the emergence of hybrid E/M phenotypes, which are yet to be validated on a single-cell level. While injection of dTAG^V pretreated EwS cells led to increased extravasation in zebrafish and lung metastasis in a mouse tail vein injection model

upon in vivo EF restoration, corroborating previous observations from a crude EF-knockdown model (Franzetti et al, 2017), cells that had undergone a 7-day EF rescue in vitro failed to recapitulate this metastatic advantage. These findings suggest that the metastatic potential conferred by transient EF fluctuation is critically dependent on the early extravasation of the highly EMT plastic hybrid cells.

The therapeutic ramifications of our findings are multifaceted. EwS is particularly sensitive to DNA-damaging agents due to EF-mediated defects in DNA double-strand break repair (Gorthi et al, 2018; Lee et al, 2020; Menon et al, 2025), and reduced EF expression may decrease responsiveness to such therapies. Metronomic treatment strategies, the continuous administration of low-dose anticancer agents currently under consideration for EwS maintenance therapy, may also be negatively affected by fluctuating EF levels, as drugs that reduce EF activity could unintentionally induce invasive, metastasis-prone hybrid states when applied at low doses. For drugs that modulate EF DNA-binding or transcriptional activity, such as cytarabine (Stegmaier et al, 2007), YK-4-279 and TK216 (Erkizan et al, 2009; Meyers et al, 2024), trabectedin (Grohar et al, 2011a), mithramycin and other mitralogues (Grohar et al, 2011b; Osgood et al, 2016), midostaurin (Boro et al, 2012), low-dose actinomycin (Chen et al, 2013b), shikonin (Chen et al, 2013a), and HCI2509 (Sankar et al, 2014) our results emphasize the need to define EF dose-dependent gene sets and phenotypes, since suboptimal inhibition may foster pro-metastatic states. Most of these compounds have been demonstrated to inhibit GGAA microsatellite-driven EF target gene activation rather than directly targeting EF protein levels, and little is known about their impact on the EF-repressive program that promotes EMT and metastasis. By contrast, molecular glues and PROTACs are being developed to directly degrade EF. Collectively, our findings underscore the necessity of completely eliminating EF activity, since incomplete inhibition may paradoxically promote metastatic progression and relapse. At the same time, we emphasize that additional validation across diverse EwS models will be required to distinguish broadly conserved from context-dependent EF-regulated pathways.

## Methods

### Reagents and tools table

| Reagent/resource | Reference or source | Identifier or catalog number |
|---|---|---|
| **Experimental models** | | |
| A673 | Cell Lines Service | RRID: CVCL_0080 |
| Lenti-X293T cells | Takara | Cat# 632180 |
| NSG mice (*NOD.Cg-Prkdc^scid Il2rg^tm1Wjl/SzJ*) | Vienna Biocenter | RRID: BCBC_4142 |
| zebrafish (*mitfa^w2/w2; mpv^a9/a*) | CCRI zebrafish facility | NA |
| zebrafish (*Tg(kdrl:Hsa.HRAS-mCherry)^s896*) | CCRI zebrafish facility | NA |
| **Recombinant DNA** | | |
| pCRIS-PITChv2-internal-mNG-dTAG | | (Brand and Winter, 2019) |

| Reagent/resource | Reference or source | Identifier or catalog number |
|---|---|---|
| pUC57 | Genscript | Cat#SD1176 |
| pRP[Exp]-EGFP/Hygro-CAG>Luc2 | VectorBuilder | Cat#VB900120-5146vmm |
| **Antibodies** | | |
| Anti-FLI1 antibody [EPR4646] | Abcam | Cat#ab133485; RRID:AB_2722650 |
| CUTANA™ IgG Negative Control Antibody | Epicypher | Cat#13-0042; RRID:AB_2923178 |
| H3K4me3 Antibody | Epicypher | Cat#13-0041; RRID:AB_3076423 |
| Acetyl-Histone H3 (Lys27) (D5E4) XP® Rabbit mAb | Cell Signaling | Cat#8173; RRID:AB_10949503 |
| Tri-Methyl-Histone H3 (Lys27) (C36B11) Rabbit mAb | Cell Signaling | Cat#9733; RRID:AB_2616029 |
| Anti-α-Tubulin Mouse mAb (DM1A) | Sigma-Aldrich | Cat#CP06; RRID:AB_2617116 |
| goat anti-rabbit DyLight 800 | Invitrogen | Cat#SA5-35571 |
| goat anti-mouse DyLight 680 | Invitrogen | Cat#35518 |
| **Oligonucleotides and other sequence-based reagents** | | |
| Oligonucleotides | see Dataset EV9 | NA |
| pCRIS-PITChv2-internal-mNG-dTAG plasmid | Georg Winter's lab (Brand and Winter, 2019) | NA |
| pUC57 | Genscript | Cat#SD1176 |
| pRP[Exp]-EGFP/Hygro-CAG>Luc2 | Oscar Martinez Tirado's lab; VectorBuilder | Cat#VB900120-5146vmm |
| **Chemicals, enzymes and other reagents** | | |
| DMEM, high glucose, GlutaMAX™ Supplement, pyruvate | Gibco | Cat#31966-021 |
| Fetal Bovine Serum | Gibco | Cat#A5256701 |
| Penicillin-Streptomycin | PAN Biotech | Cat#P06-07100 |
| Accutase | PAN Biotech | Cat#P10-21100 |
| Alt-R Cas9 Electroporation Enhancer | IDT | Cat#1075916 |
| Alt-R S.p. HiFi Cas9 Nuclease V3 | IDT | Cat#IDT1081060 |
| Alt-R HDR Enhancer | IDT | Cat#1081072 |
| M-MLV Reverse Transcriptase | Promega | Cat#M1705 |
| Oligo(dT)15 primers | Promega | Cat#C1101 |
| Q5® Hot Start High-Fidelity DNA Polymerase | New England Biolabs | Cat#M0493L |
| 1 kb DNA ladder | Promega | Cat#G5711 |
| dTAG$^V$-1 | Bio-Techne | Cat#7374 |
| Cultrex 3D Culture Matrix Rat Collagen I | Sigma-Aldrich | Cat#3447-020-01 |
| Hoechst 3342 | Invitrogen | Cat#H3570 |
| ActinRed 555 | Invitrogen | Cat#R3711 |
| Agarose, low gelling temperature | Sigma-Aldrich | Cat#A9414 |

| Reagent/resource | Reference or source | Identifier or catalog number |
|---|---|---|
| 4-thiouridine | Biosynth | Cat#NT06186 |
| Iodoacetamide | Sigma-Aldrich | Cat#I6125 |
| D-luciferin | Gold Biotechnology | Cat#LUCK-100 |
| CellTrace™ Violet | Invitrogen | Cat#C34571 |
| Tricaine | Sigma-Aldrich | Cat#E1052110G |
| **Software** | | |
| CHOPCHOP | version 3 (Labun et al, 2019) | RRID:SCR_015723 |
| FCS Express 7 | De Novo software | RRID:SCR_016431 |
| LI-COR Image Studio Software | version 5.2 | RRID:SCR_015795 |
| Fiji | ImageJ (Schindelin et al; Schneider et al, 2012) | RRID:SCR_002285 |
| Zeiss Zen 2.3 Lite | Zeiss | RRID:SCR_023747 |
| CellReporterXpress Automated Image Acquisition and Analysis Software | Molecular Devices | RRID:SCR_025681 |
| nf-core/rnaseq pipeline | version 3.14.0 | https://nf-co.re/rnaseq/3.14.0 |
| DESeq2 | version 1.42.1 | (Love et al, 2014) |
| hypeR package | version 2.0.0 | (Federico & Monti, 2020) |
| R | version 4.3.1 | https://www.r-project.org/ |
| nf-core/slamseq pipeline | version 1.0.0 | (Neumann et al, 2019) |
| nf-core/cutandrun pipeline | version 3.2.2 | nf-co.re/cutandrun/3.2.2/ |
| MACS2 | v2.1.0 | (Zhang et al, 2008) |
| ChIPseeker R package | version 1.39.0 | (Wang et al, 2022) |
| JASPAR2020 | | (Fornes et al, 2020) |
| SEURAT R package | version 4.3.0.1 | (Hao et al, 2021) |
| Living Image software | version 4.8 | RRID:SCR_014247 |
| Harmony Software | version 4.9 | |
| ComBat | | RRID:SCR_010974 |
| Graphpad prism | version 9.4.1 | RRID:SCR_002798 |
| Biorender | Science Suite Inc. | https://biorender.com/ |
| **Other kits** | | |
| MycoAlert® Mycoplasma Detection Kit | Lonza | Cat#LT07-318 |
| Gibson Assembly® Cloning Kit | NEB | Cat#E5510S |
| Guide-it long ssDNA Production System v2 | Takara | Cat#632666 |
| Cell Line NucleofectorTM Kit V | Lonza | Cat#VCA-1003 |
| RNeasy Mini Kit | Qiagen | Cat#74106 |
| CellTiter-Glo® Luminescent Cell Viability Assay | Promega | Cat#G7570 |
| Caspase-Glo® 3/7 Assay | Promega | Cat#G8091 |

| Reagent/resource | Reference or source | Identifier or catalog number |
|---|---|---|
| QuantSeq 3′mRNA-Seq V2 Library Prep Kit FWD for Illumina | Lexogen | Cat#113.96 |
| RNeasy Plus Mini Kit | Qiagen | Cat#74134 |
| Qubit RNA broad range assay | Invitrogen | Cat#Q10210 |
| PCR Add-on Kit for Illumina | Lexogen | Cat#020 |
| High-Sensitivity D1000 assay | Agilent | Cat#5067-5548; Cat#5067-5548 |
| CUTANA CUT&RUN Kit | EpiCypher | Cat#14-1048 |
| Qubit 1X dsDNA high-sensitivity assay | Invitrogen | Cat#Q33230 |
| CUTANA CUT&RUN Library Prep Kits | EpiCypher | Cat#14-1001; Cat#14-002 |

## Animal experiments and ethics

NOD.Cg-Prkdc$^{scid}$ Il2rg$^{tm1Wjl}$/SzJ (NSG) mice (Male, aged 7–12 weeks at the time of injection) were obtained from the in-house breeding facility of the Vienna Biocenter and housed in pathogen-free conditions with a housing temperature of $22 \pm 1\,°C$, $55 \pm 5\%$ humidity, and a photoperiod of 14 h light and 10 h dark. All mouse experiments were conducted in accordance with a license approved by the Austrian Ministry (GZ: MA58-2260492-2022-22). Transparent larvae from Casper zebrafish (Danio rerio) background (mitfa$^{w2/w2}$; mpv$^{a9/a9}$) stably expressing mCherry to label blood vessels (Tg(kdrl:Hsa.HRAS-mCherry)$^{s896}$) were generated from parental fish pairs bred and maintained under standard conditions at the CCRI zebrafish facility, in accordance with the license GZ:565304-2014-6 of the local authorities. All procedures were in accordance with the ARRIVE guidelines and the 3R principle.

## Cell culture

The A673 EwS cell line derived from a primary tumor localized in the muscle of a 15-year-old female patient was purchased from Cell Lines Service (CLS GmbH, Cat#300454) and authenticated by short tandem repeat (STR) profiling at Microsynth. All cells were cultured in complete DMEM medium (Gibco, Cat#31966-021) supplemented with 10% fetal calf serum (FCS, Gibco, Cat#A5256701) and 1% penicillin-streptomycin (PAN Biotech, Cat#P06-07100) at 37 °C and 5% $CO_2$. Cells were passaged at a 1:10 ratio every 3–4 days, or upon reaching confluency, using Accutase (PAN Biotech, Cat#P10-21100). Regular mycoplasma testing was performed using the MycoAlert Mycoplasma detection kit (Lonza, Cat#LT07-318).

## Plasmids and cloning materials used in this study

All plasmids and primers used in this study are summarized in Dataset EV9. To facilitate the cloning of the repair template plasmid for endogenous knock-in of A673 cells, primers were designed to amplify 500 bp homology arms corresponding to genomic sequences located immediately 5′ and 3′ of the sgRNA

cleavage sites. The mNeonGreen-HA-FKBP12$^{F36V}$ fragment was PCR amplified from the pCRIS-PITCHv2-internal-mNG-dTAG plasmid (Brand and Winter, 2019). All amplified fragments were verified for single-nucleotide polymorphisms (SNPs) by Sanger sequencing before being assembled into an EcoRV-digested pUC57 plasmid backbone (Genscript, Cat#SD1176) using the Gibson Assembly® Cloning Kit (NEB, Cat#E5510S). For in vivo mouse experiments, Luc2 expressing dual marker selection vector (pRP[Exp]-EGFP/Hygro-CAG>Luc2, VectorBuilder Cat#VB900120-5146vmm), obtained as a gift from Oscar Martinez Tirado's lab (IDIBELL Biomedical Research Institute of Bellvitge), was transfected directly into parental A673 and A2.2 cells using the TransIT-X2® Dynamic Delivery System (Mirus Bio, Cat#MIR 6004) as per the manufacturer's protocol.

## CRISPR-Cas9 genome editing of A673 cells

A CRISPR-mediated knock-in approach was employed to tag endogenous EF in A673 cells with mNG fused to HA and dTAG. An sgRNA targeting the exon-intron junction of the last exon of the FLI1 C-terminal locus was designed using the CHOPCHOP tool (Labun et al, 2019) and selected based on proximity to the FLI1 stop codon and the highest MIT specificity score. The sgRNA sequence was ordered as an Alt-R CRISPR-Cas9 sgRNA (IDT Cat#11-01-03-01, Dataset EV9). The cloned pUC57-based plasmid repair template was used to generate a single-stranded donor template with the Guide-it long ssDNA Production System v2 (Takara, Cat#632666) using the primers listed in Dataset EV9. For nucleofection, 1 million low-passage A673 cells were resuspended in nucleofector solution (Cell Line NucleofectorTM Kit V, Cat#VCA-1003) together with 2 μg of either sense or anti-sense ssDNA and 2 μL of 10.8 μM Alt-R Cas9 Electroporation Enhancer (IDT, Cat#1075916). Concurrently, an RNP complex was formed by combining Cas9 protein (Alt-R S.p. HiFi Cas9 Nuclease V3, Cat#IDT1081060) with sgRNA at a 1:1.2 molar ratio in a total volume of 10 μL with DPBS (Gibco, Cat#14190169). This mixture was incubated for 15 min at room temperature, then added to the cells for nucleofection using the Amaxa Nucleofector 2b (program x-001). Cells were then transferred to a 6-well plate containing 1 mL DMEM medium without antibiotics, supplemented with 15 μL of Alt-R HDR Enhancer (3 mM stock solution, IDT, Cat#1081072). After 24 h, the medium was replaced with a complete growth medium for cell recovery. Single cells expressing mNeonGreen were sorted by fluorescence-activated cell sorting (FACS) using a BD FACSAria™ Fusion (RRID:SCR_025715) and plated into 96-well plates.

## Validation of knock-in

Knock-in validation was initially confirmed through two-step RT-PCR using cDNA derived from the modified cells to specifically assess integration at the transcript level. Total RNA was extracted with the RNeasy Mini Kit (Qiagen, Cat#74106) according to the manufacturer's protocol, including on-column DNase digestion to remove genomic DNA contamination. First-strand cDNA synthesis was performed using 1 μg of RNA, M-MLV Reverse Transcriptase (Promega, Cat#M1705), and Oligo(dT)15 primers (Promega, Cat#C1101). PCR amplification was performed using the resulting cDNA as a template, knock-in validation primers (Dataset EV9),

and Q5® Hot Start High-Fidelity DNA Polymerase (New England Biolabs, Cat#M0493L) according to the manufacturer's guidelines. PCR products were analyzed using 0.8% TAE gel electrophoresis alongside a 1 kb DNA ladder (Promega, Cat#G5711) to confirm successful integration of the tags into the EF locus. To further validate correct integration and expression of the mNG-HA-dTAG construct at the endogenous EF locus, we conducted Western blot analysis, confocal microscopy, and flow cytometry.

## dTAG^V-1-mediated EF protein gradient

To establish an EF protein gradient, EF-dTAG clones were treated with increasing concentrations of dTAG$^V$-1 ligand (Bio-Techne, Cat#7374). Serial dilutions were prepared from a 10 mM stock to obtain micromolar working stocks, which were further diluted 1000-fold to achieve final dTAG$^V$-1 concentrations of 1.5, 5, 15, 50, and 150 nM. Cells were treated with these concentrations for different time intervals as indicated in the text. DMSO was used as a vehicle control. For ligand washout, cells were washed twice with DPBS (Gibco, Cat# 14190-094), detached using Accutase, and transferred to new culture flasks to allow full recovery of EF expression.

## Flow cytometry

Flow cytometry was employed to evaluate EF expressions upon dTAG$^V$-1 treatment for each experiment, using mNG as a readout. Cells were harvested, washed, and resuspended in Clini MACS buffer (0.5% BSA, 2 mM EDTA in DPBS). DAPI was added at a final concentration of 10 μg/mL (DAPI Staining Solution Miltenyi Biotec, Cat#130-111-570) to exclude dead cells. The fluorescence intensity of mNG was measured using a BD LSRFortessa Cell Analyzer (RRID:SCR_018655) and data analysis was performed with FCS Express 7 (De Novo software).

## Antibodies

The following primary antibodies were used for Western blot (WB) or CUT&RUN (C&R): FLI1 (Abcam Cat#ab133485, C&R 1:30), IgG (Epicypher Cat#13-0042, C&R 1:50), H3K4me3 (Epicypher Cat#13-0041, C&R 1:50), H3K27ac (Cell Signaling Technology Cat#8173, C&R 1:50), H3K27me (Cell Signaling Technology Cat#9733, C&R 1:50), Tubulin (Millipore Cat#CP06, WB 1:5,000).

## Western blot

Cells were directly lysed in 1× Laemmli buffer (2% sodium dodecyl sulfate, 10% glycerol, 5% 2-mercaptoethanol, 0.002% bromophenol blue, 0.0625 M Tris-HCl, pH 6.8) and boiled for 5 min at 95 °C. Lysates were subjected to SDS-PAGE on 8% polyacrylamide gels, and proteins were transferred onto 0.45 μm nitrocellulose membranes (Cytiva, Cat#10600002) using a transfer buffer containing 20% methanol. Membranes were blocked with Intercept blocking buffer (LI-COR, Cat#927-70001) for 1 h at room temperature and then probed with primary antibodies overnight at 4 °C. After three washes with TBS-T (200 mM Tris, 1500 mM NaCl, 0.1% Tween-20, pH 7.6), membranes were incubated with goat anti-rabbit DyLight 800 (Invitrogen, Cat#SA5-35571) or goat anti-mouse DyLight 680 (Invitrogen, Cat#35518) IgG secondary antibodies diluted 1:10,000

for 1 h at room temperature. Following three additional washes in TBS-T and rinsing with PBS, protein bands were visualized using the LI-COR Odyssey Classic Imager (RRID:SCR_023765) and LI-COR Image Studio Software version 5.2. Quantification of band intensities was performed using Fiji (Schindelin et al; Schneider et al, 2012).

## Viability assays

Cell viability and caspase activity were assessed using the CellTiter-Glo Luminescent Cell Viability Assay (Promega, Cat#G7570) and Caspase-Glo 3/7 Assay (Promega, Cat#G8091) following the manufacturer's protocols. One day prior to the experiment, cells were seeded in 96-well plates at a density of 10,000 cells per well. The following day, medium was replaced with corresponding gradient concentrations of dTAG$^V$-1 ligand. After 48 h of incubation at 37 °C and 5% $CO_2$, an equal volume of 1:4 diluted CellTiter-Glo reagent in DPBS or Caspase-Glo 3/7 reagent was added to each well. Plates were incubated at room temperature for 10 min for CellTiter-Glo and 30 min for Caspase-Glo 3/7. Luminescence was measured using an EnSpire microplate reader (PerkinElmer). All experiments were performed in triplicate, with three technical replicates per condition. S4U cytotoxicity was measured for concentrations up to 2 mM over a time scale of 8 h using the CellTiter Glo assay. S4U-containing medium was renewed after 4 h. Even at the highest concentration, no decrease in cell viability was observed (Data not shown).

## Transwell migration assays

Cells were pretreated with dTAG$^V$-1 ligand for 18 h followed by 4-h starvation in serum-free DMEM medium in the presence of dTAG$^V$-1 at indicated concentrations. Transwell inserts with a pore size of 8.0 μm (Corning, Cat#3422) were placed into 24-wells containing 600 μL complete medium. Subsequently, 200,000 cells were resuspended in 200 μL DMEM medium containing 0.1% FCS and seeded into the upper chamber. Each condition was performed in duplicate. dTAG$^V$-1 ligand was added to both the bottom and upper chamber, while DMSO served as the vehicle control. After 18–20 h of incubation at 37 °C, non-migratory cells on the upper surface of the membrane were removed using a cotton swab. Migratory cells that traversed the membrane were fixed with 4% formaldehyde solution (Sigma-Aldrich, Cat#F8775) and stained with 0.01% crystal violet in 0.25% methanol for 20 min. Images were taken at 5× magnification using a Zeiss Axio Vert series Axiovert 200 inverted microscope (RRID:SCR_020915) equipped with an AxioCam and Zeiss Zen 2.3 Lite software. Quantification was performed by counting cells in two fields of view per technical replicate using Fiji software.

## Organoplate invasion assays

Invasion assays were conducted using the Organoplate 3-lane 64 (Mimetas B.V.), which consists of 64 chips per plate, each containing three microfluidic channels (Trietsch et al, 2013). One day prior to cell seeding, Collagen I was prepared at a final concentration of 4 mg/ml (Cultrex 3D Culture Matrix Rat Collagen I, Cat#3447-020-01) and loaded into the ECM channel located in the middle lane. The plate was then incubated overnight at 37 °C

and 5% $CO_2$. The following day, cells with or without 3-week pretreatment of dTAG$^V$-1 at concentrations ranging from 0 to 150 nM were detached using Accutase, counted, and seeded at 5000 cells/μL in 2 μL of complete DMEM medium into one of the perfusion channels adjacent to the ECM channel. This resulted in passive pumping as described previously (Walker and Beebe, 2002) for a final concentration of 10,000 cells per chip. After 2 h of cell attachment, 50 μL of complete medium containing the corresponding dTAG$^V$-1 ligand concentration was added to the inlets and outlets of both perfusion channels to achieve the desired gradient concentrations. The plate was incubated at 37 °C and 5% $CO_2$ on a rocking platform (8-min intervals, 14° inclination) for 7 to 10 days. The channels are separated by two phase guides (100 × 55 μm, $w \times h$) that enables retention of ECM through pinning allowing cultured cells grown in a perfusion channel to be in direct contact with the ECM and invade through it in a barrier-free fashion (Soragni et al, 2024). At the endpoint, the plates were fixed with 3.7% formaldehyde in HBSS for 15 min. Cells were then stained with Hoechst 3342 (Invitrogen, Cat#H3570) for nuclear visualization and ActinRed 555 (Invitrogen, Cat#R3711) to stain the actin cytoskeleton. Images were acquired using the ImageXpress Confocal HT.ai (Molecular Devices) with either a 10× or 20× objective and Molecular Devices CellReporterXpress Automated Image Acquisition and Analysis Software. The images were analyzed using custom Fiji plugins developed by Mimetas. The positions of the nuclei along the Y-axis of the chip were plotted to facilitate statistical analyses of the extent of invasion.

## Soft agar colony formation assays

Cells were seeded in six-well plates with or without a 3-week pretreatment using dTAG$^V$-1 ligand at concentrations ranging from 0 to 150 nM. Following detachment with Accutase, 15,000 cells/mL were suspended in complete growth medium and mixed with 0.7% agarose in a 1:1 ratio (Sigma-Aldrich, Cat#A9414). The cell-agarose mixture was supplemented with dTAG$^V$-1 at the indicated final concentrations and then plated onto 12-well plates pre-coated with a bottom layer of 0.7% agarose. Cells were incubated at 37 °C in a humidified atmosphere with 5% $CO_2$ for 3 weeks, with medium replenished every 3–4 days to maintain dTAG$^V$-1 ligand levels. At the end of the incubation period, colonies were fixed and stained with 0.01% crystal violet in 4% formaldehyde/PBS for 2 h. Excess stains were removed by a single wash with DPBS. Colony images were captured using a flatbed scanner, and quantification was performed by manually counting the total number of colonies per well.

## RNA-sequencing

Total RNA was extracted using the RNeasy Mini Kit, following the manufacturer's instructions, including on-column DNase digestion. Quality control was performed using a DeNovix DS-11 FX spectrophotometer. Library preparation and sequencing were conducted by the Biomedical Sequencing Facility (BSF) at CeMM Research Center for Molecular Medicine of the Austrian Academy of Sciences. Libraries were prepared using the QuantSeq 3'mRNA-Seq V2 Library Prep Kit FWD for Illumina (Lexogen, Cat#113.96) and sequencing was performed on an Illumina NovaSeq 6000

Sequencing System (RRID:SCR_016387), generating 100 bp single-end reads.

## Analysis of RNA-seq data

RNA-seq reads were processed using the nf-core/rnaseq pipeline (Ewels et al, 2020) (version 3.14.0) with the GRCh38.p14 primary assembly as reference and GENCODE (RRID:SCR_014966) v46 annotation. In the pipeline, we used the following extra arguments for QuantSeq: STAR (RRID:SCR_004463) aligner options "--alignIntronMax 1000000 --alignIntronMin 20 --alignMatesGap-Max 1000000 --alignSJoverhangMin 8 --outFilterMismatchNmax 999 --outFilterMultimapNmax 20 --outFilterType BySJout --out-FilterMismatchNoverLmax 0.1 --clip3pAdapterSeq AAAAAAAA", and Salmon option "--noLengthCorrection". All data analysis was performed in R (version 4.3.1) (R core team, 2023). For differential expression analysis, we used DESeq2 (version 1.42.1, Wald test) (Love et al, 2014). We applied adaptive shrinkage to the reported fold changes using the "lfcShrink" command with type "ashr" (Stephens, 2017). We set the cutoffs for differential expression to FDR <0.05 and absolute $\log_2$FoldChange >2. To convert mapped read counts to variance-stabilized gene expression, we used the DESeq2 function "vst". For principal component analysis and visualization purposes, we averaged the expression of replicates and rescaled this expression per gene to range from 0 to 1. Principal component analysis for the 1-day-treated samples was performed using DEGs only. For the long-term-treated samples, we used all genes with a standard deviation >0.5. GSE analysis was performed using hypergeometric tests as implemented in the package hypeR (version 2.0.0) (Federico and Monti, 2020). The set of DEGs was compared against the background of all genes that could be tested (those that DESeq2 could report an adjusted $p$ value for). The gene sets tested were KEGG (RRID:SCR_012773) 2021 (Kanehisa and Goto, 2000) (only when comparing data from single-cell clones to parental bulk) and MSigDB Hallmark 2020 (Liberzon et al, 2015) as downloaded from (Federico and Monti, 2020; Kanehisa and Goto, 2000; Liberzon et al, 2015; Love et al, 2014; Stephens, 2017) https://maayanlab.cloud/Enrichr/#librarie on 10/9/2023 (Xie et al, 2021). For visualizations, we filtered the results to gene sets that had an FDR <0.001 and a log odds ratio >3 at any shown comparison. To avoid enrichment odds ratios of infinity, we imposed a weakly informative prior based on the Cauchy distribution (Gelman et al, 2008).

## SLAM-sequencing

Approximately 4.5 million cells were seeded onto 10 cm dishes and grown to 80% confluency. The next day, cells were pretreated with dTAG$^V$-1 ligand at concentrations ranging from 0 to 150 nM for 3 h to establish an EF protein gradient. Subsequently, newly synthesized RNA was labeled with 1 mM 4-thiouridine (4SU, Biosynth, Cat#NT06186) for 100 min and cells were harvested by snap-freezing plates on dry ice. RNA extraction was performed using the RNeasy Plus Mini Kit (Qiagen, Cat#74134) according to the manufacturer's instructions. Total RNA was subjected to alkylation with 10 mM iodoacetamide (IAA, Sigma-Aldrich, Cat#I6125) for 15 min, and RNA was repurified by ethanol precipitation. The purified RNA was quantified using a Qubit RNA broad range assay

(Invitrogen, Cat#Q10210). 400 ng of alkylated RNA were used as input for generating 3'-end mRNA sequencing libraries using the QuantSeq 3'mRNA-Seq Library Prep Kit FWD for Illumina (Lexogen, Cat# 113.96) and PCR Add-on Kit for Illumina (Lexogen, Cat#020). The distribution of RNA fragments was analyzed using High-Sensitivity D1000 assay (Agilent, Cat#5067-5548 and Reagents Cat# 5067-5585) in Agilent 4200 Tapestation (RRID:SCR_018435). Libraries were pooled at 4 nM, and shallow sequencing was performed using the Illumina MiSeq System (RRID:SCR_016379) to estimate the required read counts per sample, aiming for conditions where more than 3% of reads contained ≥2 T > C conversions. Deep sequencing was performed on an Illumina NovaSeq 6000 Sequencing System, generating 150 bp single-end reads.

## Analysis of SLAM-seq data

Where not stated otherwise, processing and analysis of the SLAM-seq data were performed as for the RNA-seq data. SLAM-seq reads were processed using the nf-core/slamseq pipeline (version 1.0.0) (Neumann et al, 2019). The minimum number of T > C conversions in a read to call it a T > C read was set to 2. We used the count matrix of T > C reads (newly synthesized RNA) on the transcript level for downstream analysis. Differential expression of samples with various dTAG$^V$-1 treatments compared to 0 nM dTAG$^V$-1 control was tested. Results were on the level of transcripts and to avoid multiple entries per gene, we kept the transcript with the lowest log $p$-value after averaging log $p$ values across all tests. We set the cutoffs for differential expression to FDR <0.05 and absolute log$_2$FoldChange >1. Clustering of DEGs was performed on vst-normalized and rescaled data (see RNA-seq methods for details). We used the k-means method (Hartigan and Wong, 2018) with 25 random starts for clustering. To determine the optimal number of clusters K, we varied K from 2 to 16 and calculated the gap statistic and followed the recommendations by CITE (Tibshirani et al, 2002). GSEA for clusters was performed against a background of all genes that could be tested for differential expression at any given dTAG$^V$-1 concentration.

## CUT&RUN

Cells were expanded into 10 cm dishes and pretreated with dTAG$^V$-1 ligand at concentrations ranging from 0 to 150 nM for 3 h. Following treatment, cells were detached using Accutase and subsequently frozen in aliquots of 650,000 cells per condition. The frozen cells were then subjected to the CUT&RUN protocol according to the manufacturer's instructions using the CUTANA CUT&RUN Kit (EpiCypher, Cat#14-1048). H3K4me3 and IgG antibodies served as positive and negative controls, respectively. A K-MetStat panel was added to these samples for quality control. E. coli DNA (0.25 ng) was used as a spike-in control for library normalization. After purification, CUT&RUN DNA was quantified using a Qubit 1X dsDNA high sensitivity assay (Invitrogen, Cat#Q33230). A maximum of 5 ng DNA served as input for library preparation using CUTANA CUT&RUN Library Prep Kits with primer set 1 and 2 for multiplexing (EpiCypher, Cat#14-1001 and Cat#14-002). Of note, to avoid denaturing small fragments, end repair inactivation was performed at 50 °C for 1 h instead of 65 °C for 30 min (Liu et al, 2018). The distribution of DNA fragment sizes

was analyzed using the High-Sensitivity D1000 assay in the Agilent 4200 TapeStation System. The molarity of each library was calculated and normalized to 5 nM. Libraries were pooled and sequenced using the Illumina NovaSeq 6000 Sequencing System, generating 50-bp paired-end reads.

## Analysis of CUT&RUN data

CUT&RUN reads were processed using the nf-core/cutandrun pipeline (version 3.2.2) with the GRCh38.p14 primary assembly as reference and Gencode v46 annotation. In the pipeline, we set the peak caller to MACS2 (Zhang et al, 2008) with the broad peak setting for histone marks (H3K4me3, H3K27me3, and H3K27ac) and narrow peak setting for all other samples. As a quality measure, we removed all peaks that overlapped regions that are known to cause high unspecific signal (Amemiya et al, 2019) (ENCODE4 exclusion list ENCFF356LFX). To remove spurious signal, we defined consensus peak regions per antibody target as follows: keep only significant peaks (FDR <0.01) in every replicate, create a union of peak ranges, keep only ranges that overlap significant peaks from at least two replicates. For FLI1, we combined the genotype-specific consensus peak ranges (parental and A2.2 clone) into one consensus range set by taking the union. To quantify the CUT&RUN signal at consensus peaks, we counted the fragments that map to those regions using the bamsignals R package (v 1.34.0). In subsequent analysis steps that require quantitative comparisons of conditions, the resulting peak read count matrix is analyzed with the same tools that we used for RNA-seq. To annotate the consensus peak regions with respect to their genomic location, we used the ChIPseeker (Wang et al, 2022) R package version 1.39.0 with the promoter region defined as 1kbp up- and downstream of the TSS, and the gene flanking region length set to 5kbp. We further annotated the consensus peaks with transcription factor binding motifs (TFBS) occurrence information. To do so, we used the motifmatchr R package version 1.24 to scan for all matches of the JASPAR2020 (Fornes et al, 2020) human core motifs (matchMotifs function with default $p$ value cutoff of 5e-05). To visualize the heterogeneity of the FLI1 consensus peaks based on the presence of TFBS, we transformed the high-dimensional data (12,370 peaks by 633 motifs) to a two-dimensional space. For this, we used UMAP (Healy & McInnes, 2024) with the matrix of match counts as input and the following parameters: metric = "manhattan", n_neighbors = 100, ret_nn = TRUE, pca = 10, pca_center = TRUE. To group the peaks, we calculated the shared nearest neighbor graph (Manhattan distance, 100 neighbors), pruned edges smaller than 1/15, and applied the community detection Louvain clustering.(Blondel et al, 2008) For these steps, we used implementations in the Seurat R package version 4.3.0.1 (Hao et al, 2021). To group the peaks, we calculated the shared nearest neighbor graph (Manhattan distance, 100 neighbors), pruned edges smaller than 1/15, and applied the community detection Louvain clustering (Blondel et al, 2008). For these steps, we used implementations in the SEURAT R package version 4.3.0.1 (Hao et al, 2021). We explored potential links of peak clusters with SLAM-seq gene clusters using enrichment tests. Specifically, we assigned every FLI1 consensus peak to its closest expressed gene (TSS within 150kbp) and, if the gene was in one of the five SLAM-seq clusters, assigned the gene cluster to the peak. This way, we assigned 1128 peaks to SLAM-seq clusters. Using this set of peaks,

we then tested all peak cluster and gene cluster combinations for association (two-sided Fisher's exact test).

## In vivo metastasis assay

Luciferase-expressing A2.2 EF-dTAG cells were pretreated with or without 15 nM dTAG$^V$-1 ligand for 3 weeks. An additional condition was included, where cells pretreated with 15 nM dTAG$^V$-1 for 3 weeks underwent a 7-day washout period. Following treatment, cells were detached using Accutase, resuspended in sterile PBS containing a final concentration of 15 nM dTAG$^V$-1 (for the pretreated condition), and injected intravenously via the tail vein into male NSG mice aged 7–13 weeks. A total of 1 million cells in a final volume of 100 μL were injected per mouse. Five mice were injected for each of the following conditions: A2.2 without dTAG$^V$-1 (Control), A2.2 with 15 nM dTAG$^V$-1, and A2.2 with 15 nM dTAG$^V$-1 followed by a 7-day washout (EF rescue), as well as parental A673 cells cultured for the same duration. Mice were imaged weekly using the IVIS Spectrum Xenogen bioluminescent imaging system (Caliper Life Sciences) and no further dTAG$^V$-1 ligand was supplied to the mice during the entire duration. Mice were anesthetized with 2–3% isoflurane inhalation and injected retro-orbitally with D-luciferin (150 mg/kg; Gold Biotechnology). Bioluminescent images were captured 2 min post injection to assess metastatic burden based on the bioluminescent signal (BLI). Mice were euthanized upon reaching humane endpoints (e.g., weight loss >20%, signs of distress or pain) or when the BLI signal increased by at least 100-fold compared to day 0. At the experimental endpoint, animals were anesthetized, injected with D-luciferin, and then euthanized. Lungs and livers were imaged using the IVIS system before being collected for formalin-fixed paraffin-embedded (FFPE) processing and subsequent Hematoxylin and Eosin (H&E) staining. Imaging data were analyzed using Living Image software (v.4.8).

## Zebrafish experiments

A2.2 EF-dTAG cells were pretreated with either 5 nM or 15 nM dTAG$^V$-1 ligand for 3 weeks. An additional condition involved cells that were pretreated with 15 nM dTAG$^V$-1 for 3 weeks and subsequently underwent a 7-day washout period. Following treatment, cells were detached using Accutase and stained with CellTrace™ Violet (Invitrogen, Cat#C34571) for visualization. Two-day post-fertilization (dpf) zebrafish larvae were anesthetized using 1x tricaine (pH 7.2; 0.16 g/L Ethyl-3-aminobenzoate methanesulfonate (Sigma-Aldrich, Cat#E1052110G) in E3 medium). Cells from the four different conditions (untreated, 5 nM dTAG$^V$-1, 15 nM dTAG$^V$-1, and 15 nM dTAG$^V$-1 with washout) were resuspended in PBS, with or without dTAG$^V$-1 ligand, depending on the treatment or washout conditions. To facilitate observation of the extravasation process from the tail vessels, 200–300 cells were injected into the duct of Cuvier using an Eppendorf FemtoJet 4i (RRID:SCR_019870) to directly access the circulation, as previously described (Sturtzel et al, 2023). Twenty-four xenotransplanted larvae per condition were sorted for a roughly similar cell load in circulation 2 h after injection, as observed under a fluorescence microscope. The larvae were then incubated at 34 °C without further addition of the dTAG$^V$-1 ligand to the water. Larvae were aligned in a 96-well plate (ZFA 101-02a, Funakoshi) and imaged 1- and 3-days post injection using the Operetta CLS high-content imager (RRID:SCR_018810, Revvity, formerly PerkinElmer). Images of the tail region were acquired with a 5x objective in confocal mode and evaluated using Harmony Software 4.9. This analysis located red and blue fluorescence above a set threshold to delineate the blood vessel area and define the cell area, allowing for the generation of an overlapping mask to determine the number of cells located outside or inside the vessel area.

## Data analysis, statistics, and reproducibility

All experiments were performed in at least two independent biological replicates unless stated otherwise. Animals were randomly grouped and culled prior to the planned study endpoint only when required by license stipulations or for husbandry-related reasons. Data from these individuals were excluded from further analysis. For all other experiments, no blinding or randomization was performed, and no data were excluded. The sample size ($n$) refers to biological replicates. Data were presented as the mean when $n = 2$ and as the mean ± standard error of the mean (SEM) when $n \geq 3$. Statistical analyses were conducted using one-way analysis of variance (ANOVA), two-way ANOVA with Dunnett's post hoc multiple comparisons test, as described in the figure legend for each experiment. Statistical analyses were performed using GraphPad Prism (version 9.41, GraphPad Software) and R (version 4.3.1)(R core team, 2023). Data were assumed to follow a normal distribution, except for transwell migration and soft agar assays, which were log10 or log10 + 1 transformed. Graphs were

**The paper explained**

**Problem**
Ewing sarcoma (EwS) is a highly aggressive bone cancer driven by the fusion oncoprotein EWS::FLI1 (EF). While current therapies improve outcomes for localized disease, outcomes for patients with metastatic or relapsed tumors remain poor. Despite EF's central role in tumorigenesis, the influence of fluctuations in its expression on prometastatic cell plasticity remains unclear, limiting the design of effective targeted strategies.

**Results**
Using a tunable EF expression system, we demonstrated that even transient and modest reductions in EF levels can promote a persistent migratory and invasive tumor cell state. Transcriptomic profiling revealed that EF controls distinct, dosage-dependent gene expression programs. By analyzing the chromatin architecture at EF-bound regions, we identified cofactors that cooperate with EF to regulate these programs. Importantly, we demonstrate that transient fluctuations in EF levels result in the sustained activation of genes associated with epithelial-mesenchymal plasticity.

**Impact**
This study highlights the pathobiological impact of dynamic fluctuations in oncoprotein levels on disease progression in cancers with otherwise inactive genomes, such as EwS. It suggests that therapeutic strategies targeting driver oncoproteins must completely eradicate their activity to prevent inadvertent promotion of metastasis. Furthermore, the study provides a valuable resource of genes sensitive to EF dosage and candidate co-regulators, furthering our understanding of EwS biology and revealing new therapeutic targets.

generated using GraphPad Prism (version 9.41) and R (version 4.3.1). Statistical significance was defined as *P < 0.05*.

## Graphics

The visual abstract in the Synopsis was created using tools provided by BioRender.com (Created in BioRender. Kovar, H. (2026) https://BioRender.com/sdwhfr7).

# Data availability

Raw sequencing data for this study are available at NCBI GEO under the accession numbers GSE291060 (RNA-seq), GSE291133 (SLAM-seq), and GSE291134 (CUT&RUN).

The source data of this paper are collected in the following database record: biostudies:S-SCDT-10_1038-S44321-025-00364-7.

# Peer review information

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

## Acknowledgements

We are grateful to Georg Winter and Dave Aryee for providing initial cloning constructs, and to Karin Mühlbacher for technical assistance. We thank Viktoria Humhal, Sara Wernig-Zorc, and Peter Zöscher for characterizing parental A673 and derived cell clones at the exome level. We thank Johannes Zuber's lab for helping with SLAM-seq optimization. We also acknowledge the CCRI FACS core facility for support with cell sorting and the ZANDR platform for assistance with zebrafish extravasation experiments.

This study was funded by the following grants acquired by HK: "The Austrian Science Fund (FWF) [grant-https://doi.org/10.55776/P34341 and grant-https://doi.org/10.55776/P35353], European Union's Horizon 2020 research and innovation program under the Marie Skłodowska-Curie grant agreement 956285 [VAGABOND] and Alex´s Lemonade Stand Foundation (ALSF) ["Crazy 8" grant 20-17258]. The invasion assays were supported by grants acquired by DK and KQ: Oncode Accelerator, a Dutch National Growth Fund project under grant number NGFOP2201. This work was also supported by the Cancer Grand Challenges project PROTECT to FC-A.

## Author contributions

**Veveeyan Suresh**: Data curation; Formal analysis; Investigation; Visualization; Methodology; Writing—original draft; Writing—review and editing. **Christoph Hafemeister**: Data curation; Formal analysis; Visualization; Writing—original draft; Writing—review and editing. **Andri Konstantinou**: Data curation; Investigation; Visualization; Writing—review and editing. **Sarah Grissenberger**: Data curation; Formal analysis. **Caterina Sturtzel**: Formal analysis; Visualization; Writing—review and editing. **Martha M Zylka**: Formal analysis; Investigation; Writing—review and editing. **Florencia Cidre-Aranaz**: Funding acquisition; Investigation. **Andrea Wenninger-Weinzierl**: Investigation. **Karla Queiroz**: Resources; Supervision; Funding acquisition. **Dorota Kurek**: Resources; Funding acquisition. **Martin Distel**: Resources; Supervision; Writing—review and editing. **Anna Obenauf**: Resources; Supervision. **Thomas G P Grunewald**: Resources; Supervision; Writing—review and editing. **Florian Halbritter**: Resources; Formal analysis; Supervision; Writing—review and editing. **Heinrich Kovar**: Conceptualization; Resources; Supervision; Funding acquisition; Writing—original draft; Project administration; Writing—review and editing. **Valerie Fock**: Conceptualization; Data curation; Investigation; Visualization; Methodology; Writing—original draft; Writing—review and editing.

Source data underlying figure panels in this paper may have individual authorship assigned. Where available, figure panel/source data authorship is listed in the following database record: biostudies:S-SCDT-10_1038-S44321-025-00364-7.

## Disclosure and competing interests statement

KQ and DK are employees of Mimetas, a Biotechnology company located in Oegstgeest, the Netherlands. The remaining authors declare no competing interests.

# Expanded View Figures

**Figure EV1.   Characterization of EF-dTAG clones.**

(A) Schematic depicting primer locations (arrows) on the FLI1-LHA-mNG-HA-FKBP12$^{F36V}$–FLI1-RHA homology-directed repair template used for clonal genotyping of the FLI1 locus (top). Agarose gel images showing products of two-step RT-PCR, confirming the presence of the tagged EWSR1::FLI1 fusion transcript in five EF-dTAG clones with biallelic knock-in, using the indicated primers (bottom). (B) Western blot analysis of untagged and tagged EF protein levels in parental A673 cells and EF-dTAG clones. Wild-type FLI1 (~52 kDa) was undetectable in all samples tested. Tubulin was used as a loading control. Data shown are representative of at least three independent biological replicates. (C) FLI1 CUT&RUN-seq peak signal in A2.2 dTAG clone and A673 parental line. Each dot is one of 12,370 consensus peaks. Values indicate normalized read counts, averaged across replicates. (D) Number of differentially expressed genes (DEGs) between single-cell clones and A673 parental line (left, DESeq2, FDR <0.05, absolute log$_2$ fold change >1). Functional enrichment of DEGs identifies MSigDB Hallmark 2020 and KEGG 2021 pathways as significantly enriched at least once (right, hypergeometric test, FDR <0.001, log$_2$ odds ratio >3, background: all DESeq2 tested genes). (E) Proliferation of EF-dTAG clones and parental A673 cells measured using a CellTiter-Glo assay. Data were presented as mean ± SEM ($n = 3$ independent experiments). (F) Transwell migration assays showing migrated cells stained with crystal violet 24 h after seeding, representative images from one of the three biological replicates (left). Migrated cell counts are presented as mean ± SEM after log10 transformation (right; $n = 3$), scale bar = 200 μm. (G) Organoplate invasion assays depicting cell invasion into collagen I extracellular matrix (ECM) 7 days after seeding EF-dTAG clones and parental A673. Cells were pretreated with 150 nM dTAG$^V$-1 for 24 h where indicated. Cells were stained with Hoechst (cyan, nuclei) and phalloidin (magenta, actin cytoskeleton) (left). Representative images from one of three independent biological replicates are shown, scale bars = 100 μm. Nuclei positions along the Y-axis in the organoplate invasion assays are depicted, with phase guides shown as dotted lines at distances of 535 and 930 μm (right; $n = 3$). (H) Soft agar colony formation assays showing crystal violet-stained colonies formed by EF-dTAG clones and parental A673 cells 3 weeks after seeding. Representative images from one of three independent biological replicates are shown (left). Colony counts are presented as mean ± SEM (right, $n = 3$) following log10 + 1 transformation. For panels (E–H), $P$ values were calculated using one-way ANOVA with Dunnett's post hoc multiple comparisons test. ns not significant.

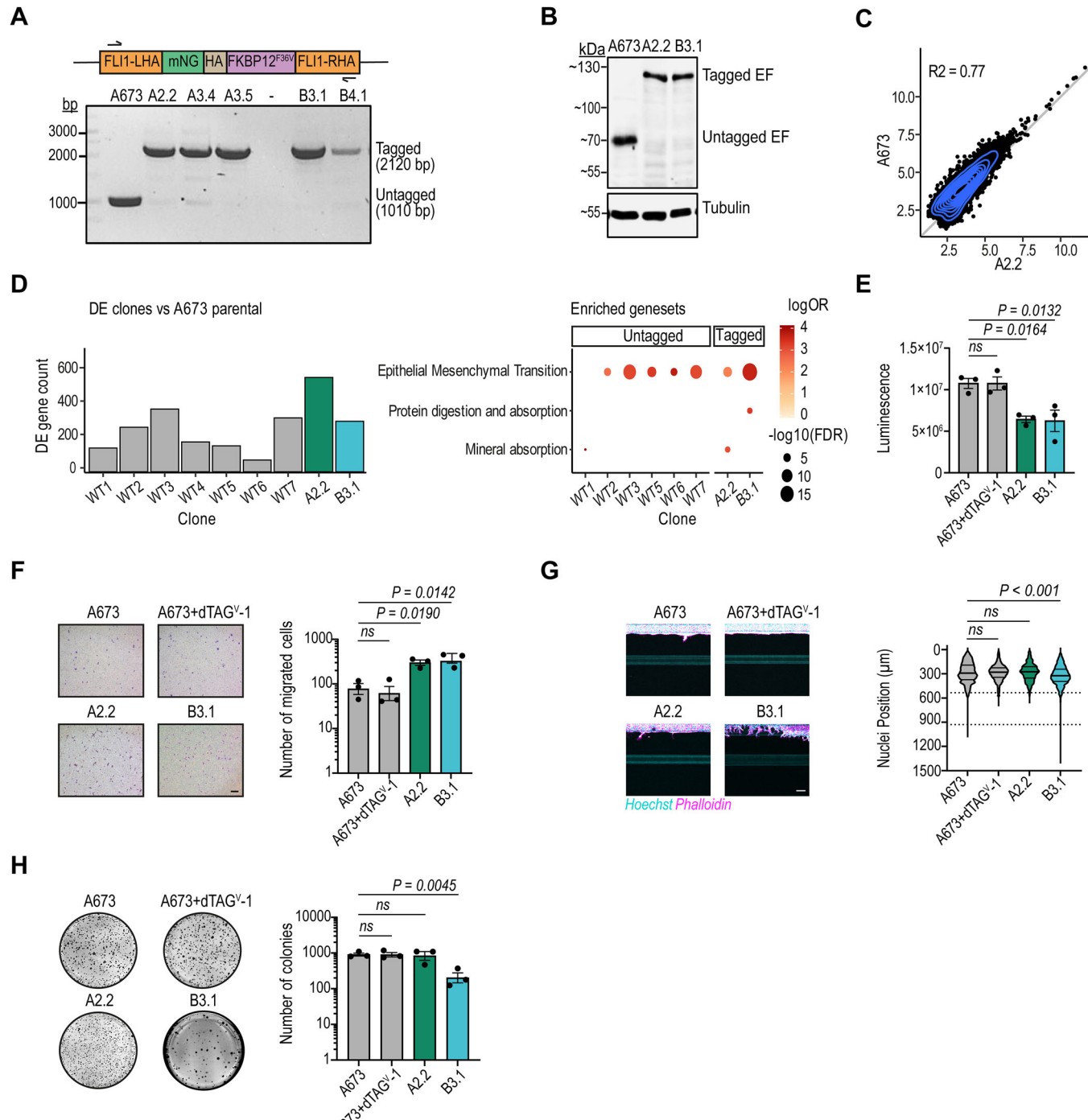

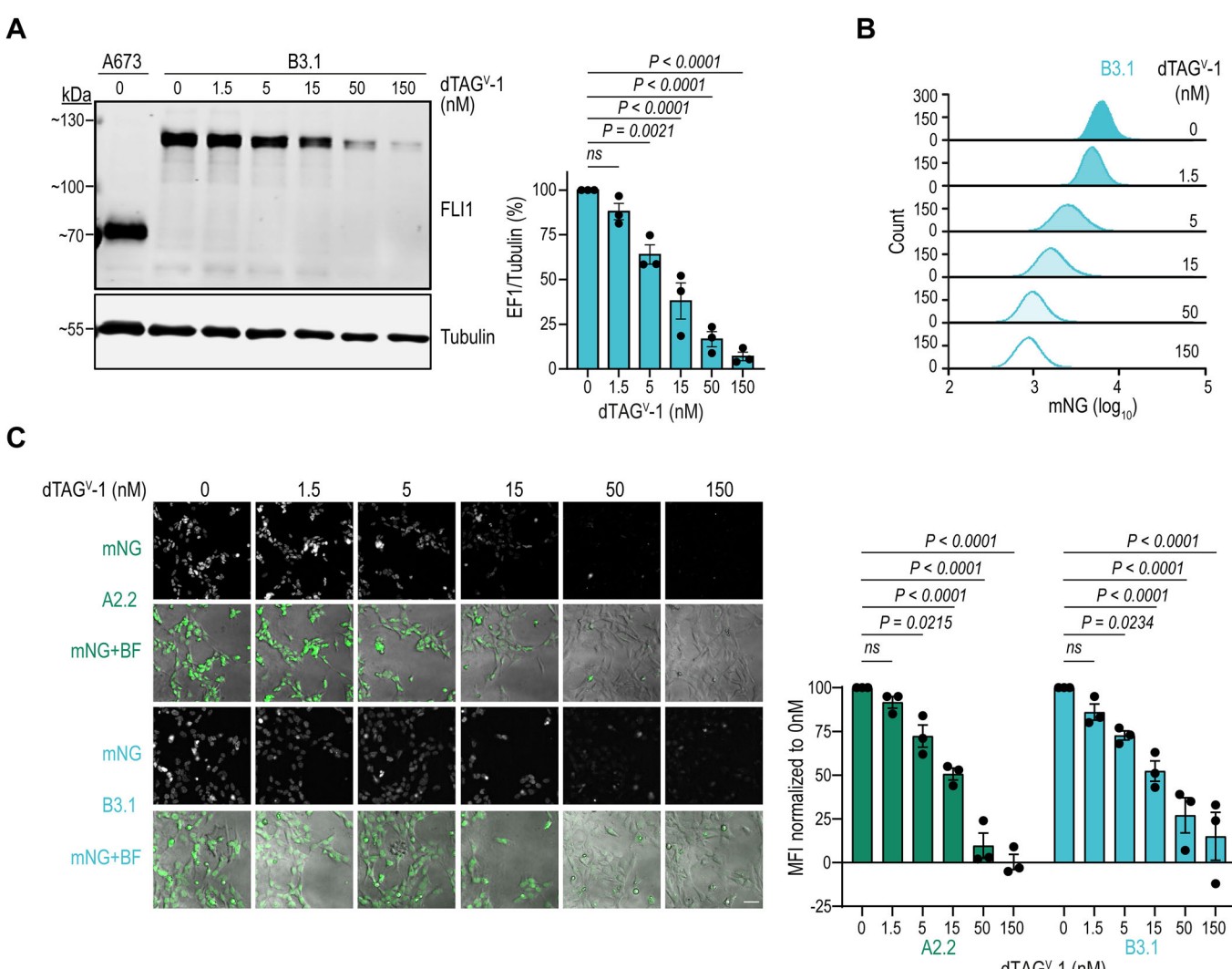

**Figure EV2. EF gradient characterization of EF-dTAG clones.**

(A) Western blot analysis of FLI1 protein levels in EF-dTAG clone B3.1 and parental A673 cells treated with the indicated concentrations of dTAG$^V$-1 for 24 h. Tubulin was used as a loading control. Bar graphs depict EF protein levels normalized to Tubulin, presented as mean ± SEM (right; $n = 3$ independent biological replicates). P values were calculated using One-way ANOVA with Dunnett's post hoc multiple comparisons. (B) Flow cytometry analysis of mNG fluorescence intensity in EF-dTAG clone B3.1 following treatment with increasing concentrations of dTAG$^V$-1 for 24 h, data were representative of at least three independent biological replicates. (C) Confocal images for mNG fluorescence in EF-dTAG clone A2.2 (top) and B3.1 (bottom) treated with the indicated concentrations of dTAG$^V$-1 for 24 h, scale bars = 50 µm. Images are representative of at least three individual experiments; BF brightfield. Bar graphs represent mean mNG mean fluorescence intensity values normalized to the 0 nM control, presented as mean ± SEM (right; $n = 3$ independent experiments). P values were calculated using two-way ANOVA with Dunnett's post hoc multiple comparisons test. ns not significant.

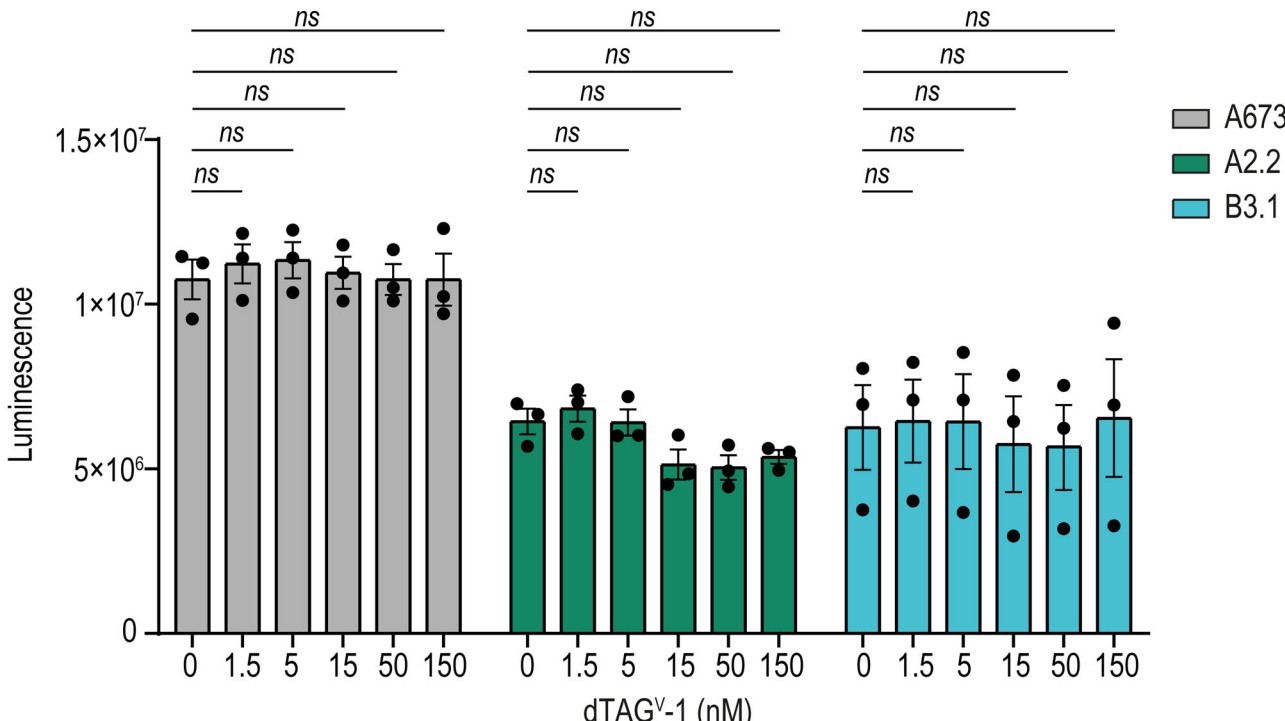

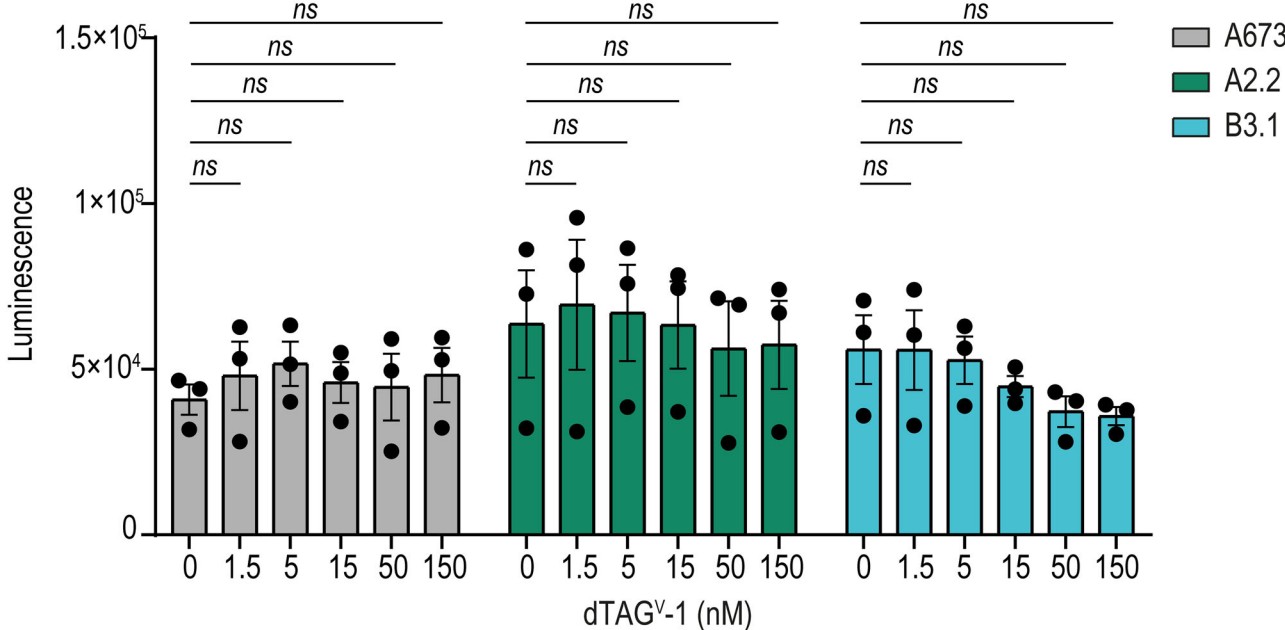

**Figure EV3. Effects of dTAG$^V$-1 on proliferation and apoptosis.**

(**A**) Proliferation of EF-dTAG clones and parental A673 cells in response to increasing concentrations of dTAG$^V$-1, assessed using the CellTiter-Glo assay. Data represents the mean ± SEM of $n = 3$ independent biological replicates. (**B**) Apoptosis of EF-dTAG clones and parental A673 cells in response to increasing concentrations of dTAG$^V$-1, measured using the Caspase-Glo 3/7 assay. Data represents the mean ± SEM of $n = 3$ independent biological replicates. $P$ values for both panels were determined using two-way ANOVA with Dunnett's multiple comparisons test. ns not significant.

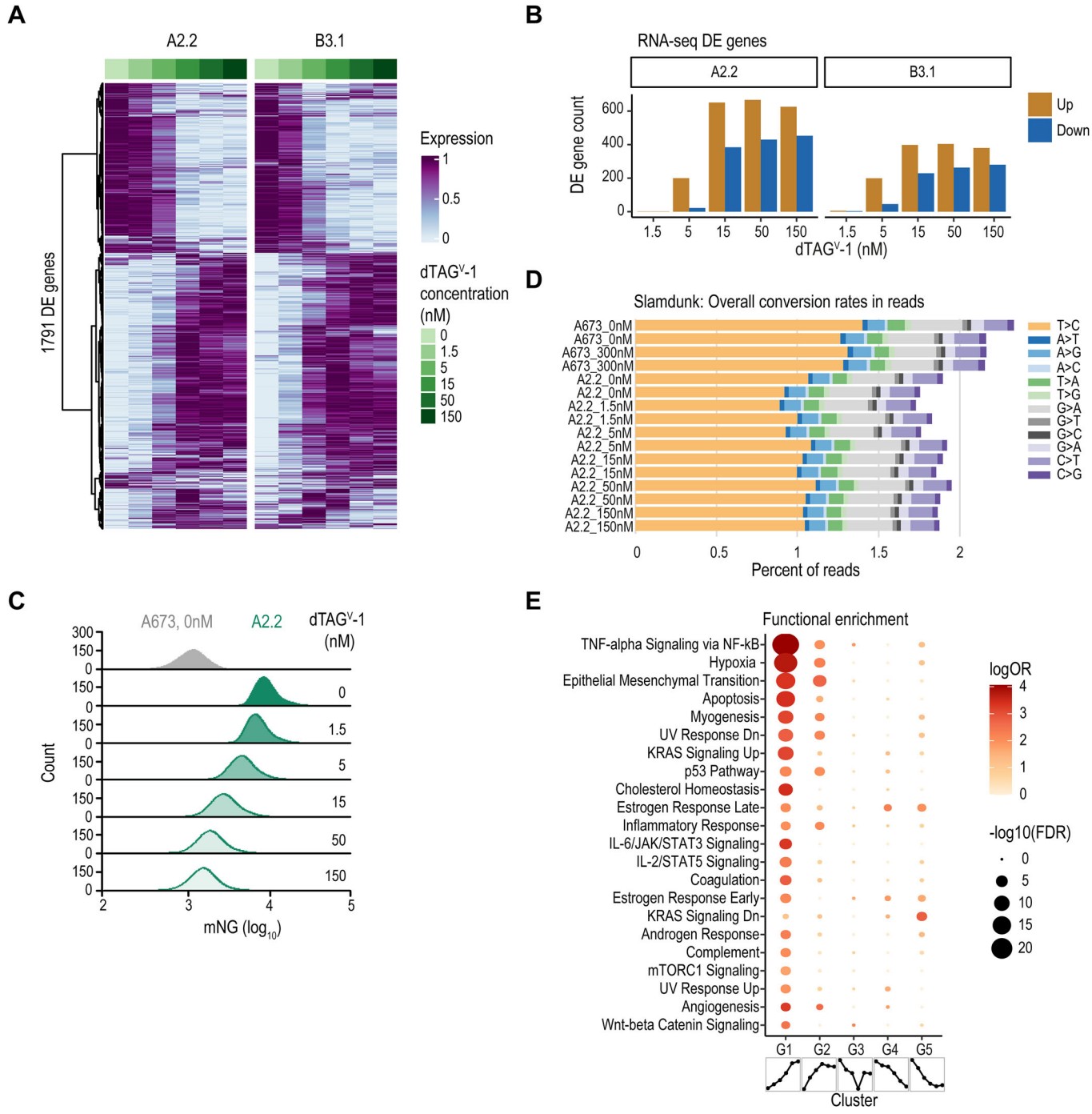

**Figure EV4.   Transcription profiling of EF-dTAG clones in response to dTAG^V-1 treatment.**

(A) Heatmaps showing gene expression changes in the 24-hour dTAG^V-1 gradient treatment experiment. The Y-axis shows all 1791 genes determined to be differentially expressed (DESeq2, FDR <0.05, absolute log$_2$ fold change >2) in at least one EF-dTAG clone at any dTAG^V-1 concentration. Gene expression was normalized using DESeq2 vst, averaged across replicates, and rescaled per gene per clone. (B) Number of DEGs from panel (A) across treatment conditions. (C) Flow cytometry analysis of mNG fluorescence intensity in EF-dTAG clone A2.2 after 3 h of treatment with increasing dTAG^V-1 concentrations, representative of samples sequenced in SLAM-seq. (D) SLAM-seq base conversion rates in percent of all reads as reported by the Slamdunk processing pipeline. (E) Functional enrichment analysis of SLAM-seq cluster genes, showing MSigDB Hallmark 2020 pathways that were significantly enriched (hypergeometric test, FDR <0.05, log$_2$ odds ratio >1, background: all detected genes) in at least one cluster. Icons below the cluster labels show the mean expression pattern per cluster (x-axis shows increasing dTAG^V-1 concentrations, y-axis rescaled expression).

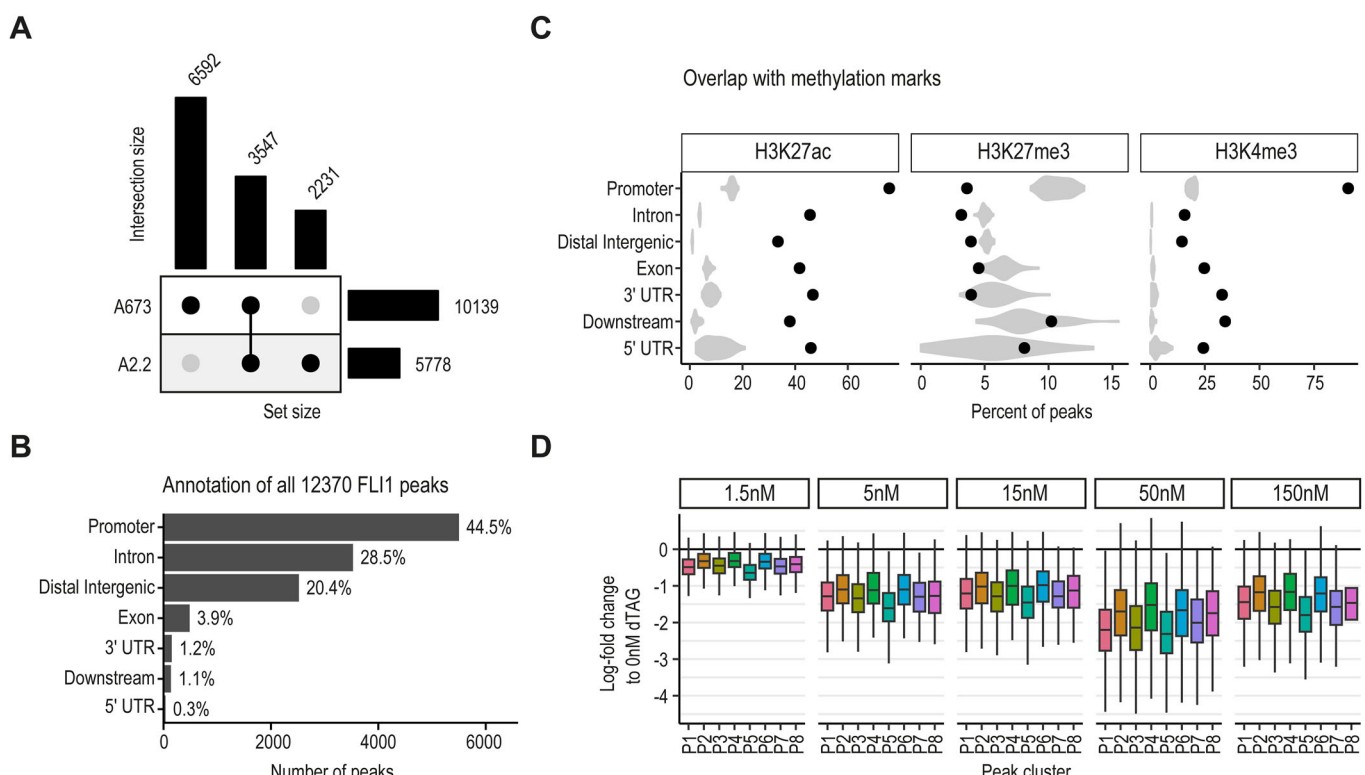

**Figure EV5.   Characterization of FLI1-binding peaks and dynamics of peak signals.**

(**A**) UpSet plot showing the intersection of the CUT&RUN FLI1 peaks discovered in A2.2 dTAG clone and parental A673 cells. (**B**) Annotation of FLI1 peaks with respect to genomic location (Promoter = 5501, Intron = 3527, Distal intergenic = 2520, Exon = 485, 3'UTR = 152, Downstream = 137, 5'UTR = 37). (**C**) Percentage of FLI1 peaks (x-axis), grouped by genomic location (y-axis), overlapping methylation marks. Observed percentages are shown as black points and random background distributions of shuffled peaks (all peaks randomly placed in the genome 50 times) depicted as gray violin plots. (**D**) FLI1 CUT&RUN peak signal change for all FLI1 consensus peaks grouped by peak cluster and dTAG$^V$-1 concentration in A2.2 clone. Boxplots show log-fold change compared to 0 nM condition as reported by DESeq2 with ashr shrinkage applied (P1 = 4160, P2 = 2409, P3 = 2079, P4 = 1367, P5 = 958, P6 = 661, P7 = 538, P8 = 198). Boxplots show 25% (lower hinge), 50% (horizontal line), and 75% (upper hinge) quantiles. The whiskers extend from the lower/upper hinge to the smallest/largest value within 1.5 times the hinge spread. Outliers are not shown.

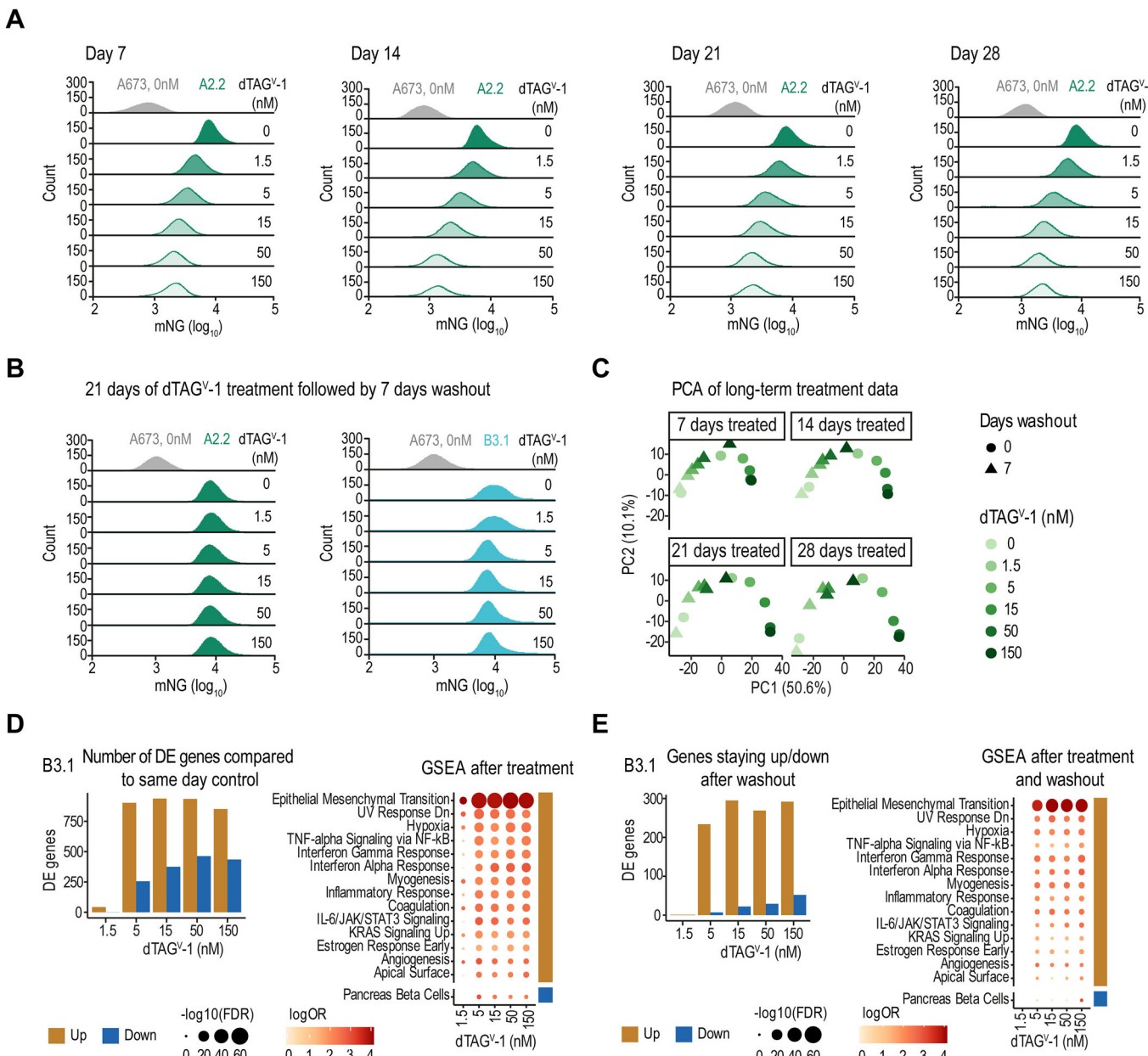

**Figure EV6. Long-term EF gradient treatment and transient effects due to dTAG^V-1 treatment.**

(A) Flow cytometry analysis of mNG fluorescence intensity in EF-dTAG clone A2.2 after 7, 14, 21, and 28 days of treatment with increasing concentrations of dTAG^V-1. Representative data corresponds to one of the technical replicates used for RNA-sequencing. (B) Flow cytometry analysis of mNG fluorescence intensity in EF-dTAG clones after 21 days of dTAG^V-1 treatment at the indicated concentrations, followed by a 7-day washout period. Representative data corresponds to one of the technical replicates used for RNA-sequencing. (C) PCA of prolonged and transient EF dosage modulation on transcription in A2.2 clone (RNA-seq), showing the first two principal components calculated using 1455 highly variable genes (gene expression with standard deviation >0.5 after normalization with DESeq2 vst function and averaging two replicates). (D) Number of DEGs (Left, DESeq2, FDR <0.05, absolute log_2 fold change >2) and functional pathways enrichment analysis (Right) for clone B3.1 after 21-day treatment with increasing concentrations of dTAG^V-1. Pathways shown are MSigDB Hallmark 2020 pathways that were significantly enriched (hypergeometric test, FDR <0.001, log_2 odds ratio >3) in any treatment condition. (E) As (D) but showing results for treatment followed by a 7-day washout.

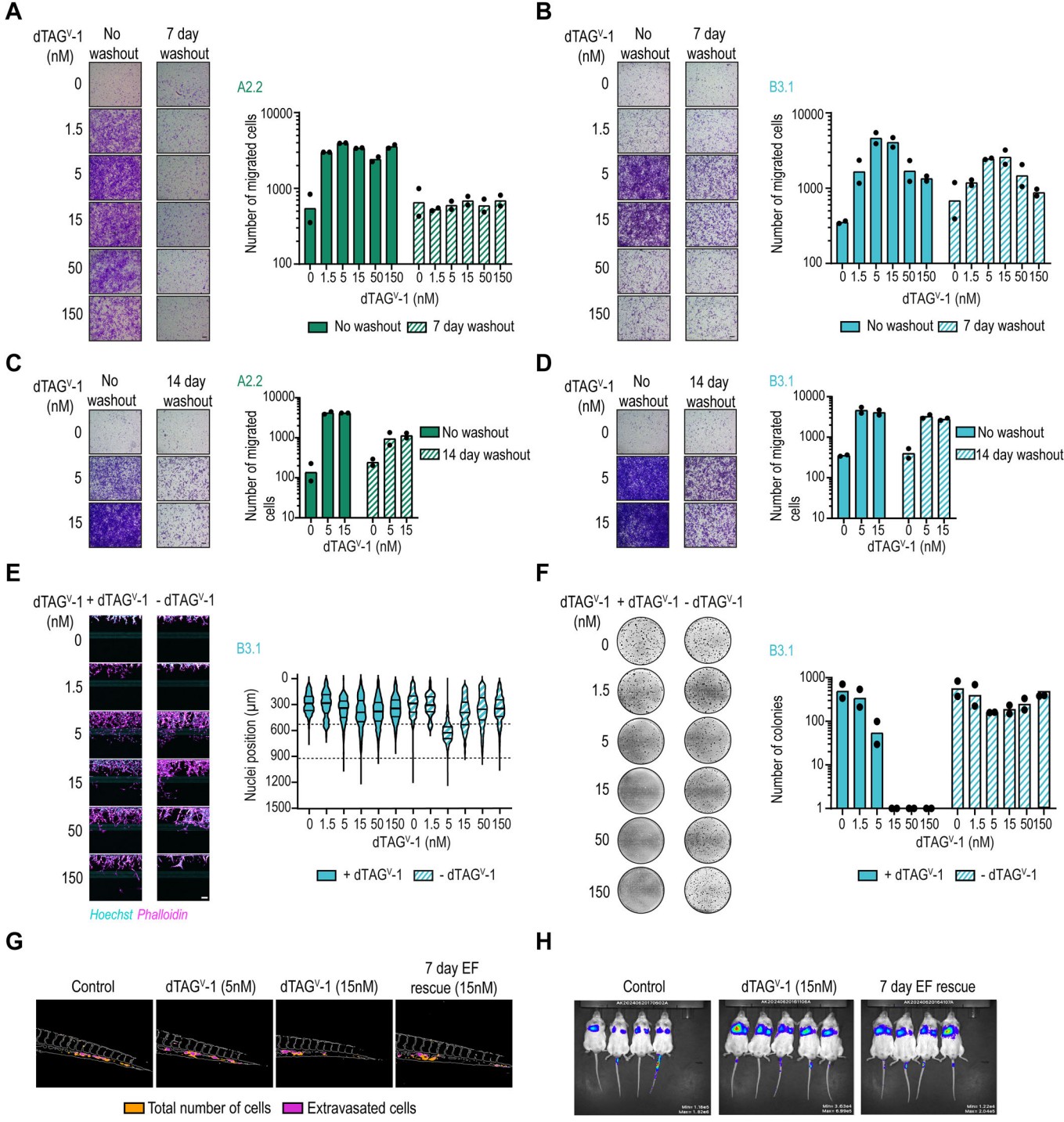

◀ **Figure EV7. Metastatic phenotypic analysis of EF-dTAG clones following 21-day dTAG$^V$-1 treatment and washout.**

(A) Transwell migration assays showing migrated A2.2 cells after 7-day treatment with dTAG$^V$-1 at the indicated concentrations or treatment for 7 days followed by a 7-day washout. Representative images from one of two independent biological replicates are shown (left). Migrated cell counts are presented as mean after log10 transformation (right; $n = 2$). Scale bar $= 200$ μm. (B) Transwell migration assays showing migrated B3.1 cells following treatment with dTAG$^V$-1 for 21 days at the indicated concentrations, or treatment for 21 days followed by a 7-day washout. Representative images from one of two independent biological replicates are shown (left). Migrated cell counts, presented as mean after log10 transformation, are shown on the right ($n = 2$). Scale bar $= 200$ μm. (C, D) Transwell migration assays showing migrated A2.2 cells (C) or B3.1 cells (D) following 21-day treatment with dTAG$^V$-1 at the indicated concentrations as in Fig. 6A (A2.2 no washout) and EV7B (B3.1 no washout) or following 21 days of treatment and a subsequent 14-day washout. Representative images from one of two independent biological replicates are shown (left). Migrated cell counts are presented as mean after log10 transformation (right; $n = 2$). Scale bar $= 200$ μm. (E) Organoplate invasion assays showing cell invasion into collagen I ECM 7 days after seeding B3.1 cells treated with dTAG$^V$-1 for 21 days at the indicated concentrations, with or without further dTAG$^V$-1 treatment. Representative images from one of two independent biological replicates are shown (left). Mean nuclei positions along the Y-axis are shown, with phase guides indicated by dotted lines at 535 and 930 μm (right; $n = 2$). Scale bar $= 100$ μm. (F) Soft agar assays showing colonies formed by B3.1 cells three weeks after seeding 21 days dTAG$^V$-1 treated cells at indicated concentrations with or without further dTAG$^V$-1 treatment. Representative images from one of two independent biological replicates are shown (left). Colony counts are presented as mean ($n = 2$) following log10 $+$ 1 transformation. (G) Representative images of zebrafish extravasation assays 3 days post injection of cells treated with dTAG$^V$-1 for 21 days at the indicated concentrations, or 21 days of 15 nM dTAG$^V$-1 followed by a 7-day washout. (H) Representative IVIS images of mice showing the homing of luciferase-expressing A2.2 cells to lungs directly after (Day 0) tail vein injection of cells treated with 15 nM dTAG$^V$-1 for 21 days, with or without a 7-day washout, along with a control. $P$ values were calculated using two-way ANOVA with Dunnett's post hoc multiple comparisons test. ns not significant.

