## [Peer Review File · EMBO Molecular Medicine]

Modelling EWS::FLI1 protein fluctuations reveal determinants of tumor plasticity in Ewing sarcoma

Veveeyan Suresh, Christoph Hafemeister, Andri Konstantinou, Sarah Grissenberger, Caterina Sturtzel, Martha Zylka, Florencia Cidre-Aranaz, Andrea Wenninger-Weinzierl, Karla Queiroz, Dorota Kurek, Martin Distel, Anna Obenauf, Thomas Grunewald, Florian Halbritter, Heinrich Kovar, and Valerie Fock

Corresponding authors: Heinrich Kovar (heinrich.kovar@ccri.at) , Valerie Fock (valerie.fock@outlook.com)

Review Timeline:

Submission Date:	26th Jun 25
Editorial Decision:	29th Jul 25
Revision Received:	23rd Oct 25
Editorial Decision:	3rd Nov 25
Revision Received:	14th Nov 25
Accepted:	18th Nov 25

Editor: Lise Roth

Transaction Report:

29th Jul 2025

Dear Prof. Kovar,

Thank you for submitting your manuscript to EMBO Molecular Medicine, and please accept my apologies for the delay in getting back to you as one referee needed more time to complete their review. We have now received feedback from the three reviewers who agreed to evaluate your manuscript. As you will see from the reports below, they acknowledge the interest of the study and are overall supporting publication of your work pending appropriate revisions.

Following further consultation with the referees, we agreed that functional validation assays would NOT be required, but that the referees' concerns should be addressed through robust adjustments to the text. Please note that Referee #3 also agreed with Referee #2 that Figure 7 should be removed.

Addressing the reviewers' concerns in full will be necessary for further considering the manuscript in our journal, and acceptance of the manuscript will entail a second round of review. EMBO Molecular Medicine encourages a single round of revision only and therefore, acceptance or rejection of the manuscript will depend on the completeness of your responses included in the next, final version of the manuscript. For this reason, and to save you from any frustrations in the end, I would strongly advise against returning an incomplete revision.

We are expecting your revised manuscript within three months, if you anticipate any delay, please contact us.

We require:

- 1) A .docx formatted version of the manuscript text (including legends for main figures, EV figures and tables). Please make sure that the changes are highlighted to be clearly visible.
- 2) Individual production quality figure files as .eps, .tif, .jpg (one file per figure). For guidance, download the 'Figure Guide PDF' (<https://www.embopress.org/page/journal/17574684/authorguide#figureformat>).
- 3) At EMBO Press we ask authors to provide source data for the main manuscript figures. You will receive a separate email with instructions for providing source data with your revised manuscript, including how to upload and organize the files.

Additional information on source data and instruction on how to label the files are available

- 4) A .docx formatted letter INCLUDING the reviewers' reports and your detailed point-by-point responses to their comments. As part of the EMBO Press transparent editorial process, the point-by-point response is part of the Review Process File (RPF), which will be published alongside your paper.

- 5) A complete author checklist, which you can download from our author guidelines (<https://www.embopress.org/page/journal/17574684/authorguide#submissionofrevisions>). Please insert information in the checklist that is also reflected in the manuscript. The completed author checklist will also be part of the RPF.

- 6) All Materials and Methods need to be described in the main text using our 'Structured Methods' format. According to this format, the Methods section includes a Reagents and Tools Table (listing key reagents, experimental models, software and relevant equipment and including their sources and relevant identifiers) followed by a Methods and Protocols section describing the methods, ideally using a step-by-step protocol format. The aim is to facilitate adoption of the methodologies across labs. Please download and fill our Reagents and Tools Table template (.docx), which you can find in our author guidelines: <https://www.embopress.org/page/journal/14693178/authorguide#structuredmethods>.

- 7) Please note that all corresponding authors are required to supply an ORCID ID for their name upon submission of a revised manuscript.

- 8) It is mandatory to include a 'Data Availability' section after the Materials and Methods. Before submitting your revision, primary datasets produced in this study need to be deposited in an appropriate public database, and the accession numbers and

database listed under 'Data Availability'. Please remember to provide a reviewer password if the datasets are not yet public (see <https://www.embopress.org/page/journal/17574684/authorguide#dataavailability>).

9) For data quantification: please specify the name of the statistical test used to generate error bars and P values, the number (n) of independent experiments (specify technical or biological replicates) underlying each data point and the test used to calculate p-values in each figure legend. The figure legends should contain a basic description of n, P and the test applied. Graphs must include a description of the bars and the error bars (s.d., s.e.m.). Please provide exact p values.

10) Our journal encourages inclusion of *data citations in the reference list* to directly cite datasets that were re-used and obtained from public databases. Data citations in the article text are distinct from normal bibliographical citations and should directly link to the database records from which the data can be accessed. In the main text, data citations are formatted as follows: "Data ref: Smith et al, 2001" or "Data ref: NCBI Sequence Read Archive PRJNA342805, 2017". In the Reference list, data citations must be labeled with "[DATASET]". A data reference must provide the database name, accession number/identifiers and a resolvable link to the landing page from which the data can be accessed at the end of the reference. Further instructions are available at .

11) We replaced Supplementary Information with Expanded View (EV) Figures and Tables that are collapsible/expandable online. EV Figures should be cited as 'Figure EV1, Figure EV2' etc... in the text and their respective legends should be included in the main text after the legends of regular figures.

12) The paper explained: EMBO Molecular Medicine articles are accompanied by a summary of the articles to emphasize the major findings in the paper and their medical implications for the non-specialist reader. Please provide a draft summary of your article highlighting

13) Author contributions: CRediT has replaced the traditional author contributions section because it offers a systematic machine readable author contributions format that allows for more effective research assessment. Please remove the Authors Contributions from the manuscript and use the free text boxes beneath each contributing author's name in our system to add specific details on the author's contribution. More information is available in our guide to authors.

Please also suggest a visual abstract to illustrate your article as a PNG file 550 px wide x 300-600 px high. A cropped portion of this image will serve as thumbnail for the table of content on our webpage.

16) As part of the EMBO Publications transparent editorial process initiative (see our Editorial at <http://embomolmed.embopress.org/content/2/9/329>), EMBO Molecular Medicine will publish online a Review Process File (RPF) to accompany accepted manuscripts.

In the event of acceptance, this file will be published in conjunction with your paper and will include the anonymous referee reports, your point-by-point response and all pertinent correspondence relating to the manuscript. Let us know whether you

agree with the publication of the RPF and as here, if you want to remove or not any figures from it prior to publication. Please note that the Authors checklist will be published at the end of the RPF.

I look forward to receiving your revised manuscript.

Yours sincerely,

Lise Roth

***** Reviewer's comments *****

Referee #1 (Remarks for Author):

This is a very interesting demonstration of the importance of eF dosage in the phenotype of EWS cells. The authors have build convincing models and identified the genes over/repressed in the different concentrations of the EF opening interesting hypothesis on the mechanisms of resistance to cytotoxic treatment in vivo. I have several questions:

- 1) the constructions enabling the modulation of EF dosage were not found working in all EWS cells lines, and the authors mention this as "context dependent". What is the hypothesis behind this?
- 2) what is the impact of EF dose in response to cytotoxics, (including those cited as modulating EF binding) or radiation therapies?
- 3) have the authors been able to study sequential samples in patients, in terms of expression profiles , and EF dosages at different time point of disease course.
- 4) have the authors tested the impact of further overexpression of EF in A673 cells in terms of phenotype and gene expression.

Referee #2 (Comments on Novelty/Model System for Author):

All experiments were carried out in one high-passage cell line. The authors evaluated two clones and executed careful control experiments. Nevertheless, the fact that they were not able to reproduce their results in another line is somewhat concerning and puts the general applicability of their findings in question.

Referee #2 (Remarks for Author):

Suresh, Hafemeister et al use degron technology to titrate the dose of EWS::FLI1 in the high-passage EWS cell line A673. Reduced EWS::FLI1 dose correlated with increased cell migration and reduced anchorage-independent growth. Proliferation and apoptosis did not change. Clusters of genes, whose expression appeared to respond differentially to changes in EWS::FLI1 dose, were identified. Enhancers containing GGAA microsatellites and Krüppel-like zinc finger transcription factors were identified as candidate co-regulators of genes with differential expression in response to EWS::FLI1 dose.

I strongly believe that plasticity in the expression of sarcoma-driving fusion oncogenes is a major factor in the malignancy of these cancers and has not received sufficient attention thus far. Any project addressing these issues comes with major challenges. I applaud the authors for embarking on this journey and urge the editorial team to consider the general importance of this topic.

Major concerns about the study are as follows:

- Obviously, the major weakness of this paper is that all experiments were carried out in one high-passage cell line. The authors

evaluated two clones and executed careful control experiments. Nevertheless, the fact that they were not able to reproduce their results in another line is somewhat concerning and puts the general applicability of their findings in question. Are the conclusions from this study truly representative for Ewing sarcomas? This limitation should be stated very clearly in the manuscript.

- The study includes extensive bioinformatic analyses to support the notion that enhancers containing GGAA microsatellites and Krüppel-like zinc finger transcription factors act as co-regulators of genes with differential expression in response to EWS::FLI1 dose. Experimental evidence to validate the functional relevance of these observations would strengthen the manuscript substantially.

- Another relevant weakness is that previous studies indicated that differences in the expression of EWS::FLI1 were dynamic over time and truly plastic. How do the authors reconcile these published findings with persistent dysregulation of certain genes after transient reduction of EWS::FLI1 dose? The authors need to discuss the discrepancy of their findings with published literature.

- The authors show that expression of 23 out of 85 genes (associated with EMT and reduced EWS::FLI1 dose) correlated with survival. I am sorry, but I think this correlation is fairly random and does not enhance the relevance of the main findings reported in this paper. I would consider removing figure 7 from the manuscript.

In addition, I would like to recommend further discussion of the concept of plasticity. The authors emphasize that therapies aimed at eliminating EWS::FLI1 need to completely eradicate its activity and must avoid pro-tumorigenic effects at lower EWS::FLI1 doses. However, EWS::FLI1 plasticity may have important implications that go far beyond this statement in terms of development of resistance and metronomic therapy. This should be discussed.

If the authors can address the issues outlined above and clearly discuss the limitations of the study, I recommend to consider accepting this paper in light of the fact that it sheds further light on complex and underappreciated aspect of Ewing sarcoma biology.

Referee #3 (Comments on Novelty/Model System for Author):

The manuscript provides novel candidate mechanistic evidence for disease relevant processes in EWS, including metastasis and the E/M and through this generates clear value. However it relies on the use of a single Ewing sarcoma (EWS) cell line (A567), and functional gene editing could not be achieved in other EWS cell lines. This restricts the generalizability of the findings.

Referee #3 (Remarks for Author):

The manuscript seeks to examine the effect of fluctuation in the level of EWS/FLI (EF) fusion oncogene, which are known to arise in EWS and to promote metastatic activity of EWS. Using an elegant strategy to edit the endogenous fusion gene with a degron c-terminal tag the authors undertake high end transcriptome characterisation and alongside phenotypic response monitoring in Ewing sarcoma A567 cells.

Experiments are very well designed and controlled and data presented are convincing. Overall the manuscript confirms existing evidence concerning the response of EWS oncogenic signalling to changes in the dose of EF fusion oncogene product and extends knowledge of the transcriptional regulatory landscape involved in process. Amongst others the work raises novel hypothesis concerning the involvement of cellular transcription factors that may deliver EF signalling output.

The study has several limitations that should be clearly acknowledged. First, it relies on the use of a single Ewing sarcoma (EWS) cell line (A567), and functional gene editing could not be achieved in other EWS cell lines. This restricts the generalizability of the findings and raises questions about whether the observed regulatory networks are broadly relevant across EWS subtypes.

Moreover, the inability to edit EF in other cell lines suggests that certain molecular functions of EF may be essential in those contexts but are dispensable in A567. This functional variability limits the ability to model and safely predict the general or principal impact of EF-targeting therapeutics. In other words, potential EF functions that are not evident in A567 may counteract or overshadow the risk-such as enhanced metastasis-identified in this study. Therefore, any claims regarding EF-targeted therapies should be moderated and carefully weighed against this context.

Finally, the conclusions drawn in the study are largely based on correlative data and have not been experimentally validated. Additional functional validation will be essential in future work. These limitations should be more explicitly discussed in the manuscript's discussion section to provide a balanced and accurate interpretation of the findings.

In conclusion and overall the manuscript provides novel candidate mechanistic evidence for disease relevant processes in EWS, including metastasis and the E/M. Some cautionary text revision as outlined above should be considered by the authors.

Point-by-point response to reviewer comments:**Referee #1:**

This is a very interesting demonstration of the importance of EF dosage in the phenotype of EWS cells. The authors have built convincing models and identified the genes over/repressed in the different concentrations of the EF opening interesting hypothesis on the mechanisms of resistance to cytotoxic treatment in vivo.

We thank the reviewer for this positive assessment and constructive questions.

Concern: *the constructions enabling the modulation of EF dosage were not found working in all EWS cells lines, and the authors mention this as "context dependent". What is the hypothesis behind this?*

Our response: We thank the reviewer for highlighting this important point. In agreement with a recent preprint reporting auxin tagged EF (McGinnis *et al*, 2024), our data also indicate that larger C-terminal fusions may affect aspects of *EWSR1::FLI1* function, resulting in a hypomorphic allele. In lines where the model was unsuccessful (TC71 and STA-ET 7.2), we obtained clones that had duplicated their *EWSR1::FLI1* locus, leading to co-expression of a non-degradable wildtype EF alongside the modified variant (now in Appendix Fig. S1A and referenced in the Results section line 93). This is substantiated by studies showing that cancer cells may respond to potentially detrimental gene editing at essential loci by amplifying or duplicating the targeted region, thus safeguarding critical protein function (Aguirre *et al*, 2016).

Appendix Fig. S1A: C-terminal EF tagging leads to *EF* gene duplication in TC71 and STA-ET-7.2 cells.

(A) Western blot analysis showing protein levels of endogenous untagged EF (~70 kDa) and dTAG-fused (~130 kDa) EF in parental (TC71, STA-ET-7.2) and EF-dTAG clones (TC71-A6, A15; STA-ET-7.2-C18, C19). Tubulin was used as a loading control.

While neither differences in the mutation status of the most recurrently mutated genes in ES (*TP53*, *STAG2*, *CDKN2A*), nor differences in EF fusion type explain distinct tolerance to C-terminal EF modification, we speculate that this may be due to the presence of a *BRAFV600E* mutation that is specifically present in the A673 cell line. Our data supports the view that tolerance (or intolerance) to C-terminal EF editing is highly dependent on the cell's genomic and/or epigenomic landscape. This likely reflects adaptive responses to preserve EF function, shaped by each line's unique (epi-)genetic context. To clarify this, we have rephrased the sentence in lines 277-281 to read:

“However, our unsuccessful attempts to establish a similar model in other EwS cell lines suggest that tolerance to carboxy-terminal EF modification may depend on the underlying genomic or epigenomic background reflecting a cellular adaptive response to maintain EF function, likely shaped by each line's unique (epi-)genetic context. The precise molecular mechanisms for this selectivity remain to be determined. Therefore, the restriction to a single cell line has to be seen as the major limitation of our study.”

Concern: What is the impact of EF dose in response to cytotoxics, (including those cited as modulating EF binding) or radiation therapies?

Our response: This is indeed a clinically highly relevant question. Our data implies multifaceted consequences of altered EF levels.

Mechanistically, EF directly binds to DNA double-strand breaks and perturbs multiple branches of resection-dependent DNA damage repair. It does so by replacing EWS from the DNA breaks, suppressing homologous recombination through R-loop-mediated BRCA1 interference (Gorthi *et al*, 2018), promoting PARP1 immobilization on chromatin (Lee *et al*, 2020), and causing

functional ATM deficiency (Menon *et al*, 2025). Based on these mechanisms, one would predict that decreased EF levels would reduce sensitivity to DNA-damaging cytotoxics, although this has not yet been directly examined under fluctuating EF states.

In addition, we propose that transient EF fluctuations (> 3 weeks) may stabilize a hybrid epithelial/mesenchymal like phenotype. Such states are strongly associated with therapy resistance and poor survival across cancers (Biddle *et al*, 2016; Fustaino *et al*, 2017; Hanrahan *et al*, 2017; Wei *et al*, 2025). In breast cancer, for example, the frequency of hybrid states change during systemic therapy and progression (Yu *et al*, 2013). We speculate that similar plasticity dynamics contributing to intra-tumoral heterogeneity may occur during metronomic treatment, i.e., the continuous administration of low dose anticancer agents currently discussed for EwS maintenance therapy. Agents that partially reduce EF activity may therefore unintentionally induce invasive, therapy-resistant phenotypes. Accordingly, drugs targeting EF levels or function may not be suited for metronomic/low-dose regimens. We have revised the Discussion (lines 402-419) to critically reflect these aspects.

“The therapeutic ramifications of our findings are multifaceted. EwS is particularly sensitive to DNA-damaging agents due to EF-mediated defects in DNA double-strand break repair (Gorthi *et al.*, 2018; Lee *et al.*, 2020; Menon *et al*, 2025), and reduced EF expression may decrease responsiveness to such therapies. Metronomic treatment strategies, the continuous administration of low-dose anticancer agents currently under consideration for EwS maintenance therapy, may also be negatively affected by fluctuating EF levels, as drugs that reduce EF activity could unintentionally induce invasive, metastasis-prone hybrid states when applied at low dose. For drugs that modulate EF DNA-binding or transcriptional activity, such as cytarabine (Stegmaier *et al*, 2007), YK-4-279 and TK216 (Erkizan *et al*, 2009; Meyers *et al*, 2024), trabectedin (Grohar *et al*, 2011a), mithramycin and other mitralogues (Grohar *et al*, 2011b; Osgood *et al*, 2016), midostaurin (Boro *et al.*, 2012), low-dose actinomycin (Chen *et al*, 2013b), shikonin (Chen *et al*, 2013a), and HCI2509 (Sankar *et al.*, 2014) our results emphasize the need to define EF dose-dependent gene sets and phenotypes, since suboptimal inhibition may foster pro-metastatic states. Most of these compounds have been demonstrated to inhibit GGAA microsatellite-driven EF target gene activation rather than directly targeting EF protein levels, and little is known about their impact on the EF-repressive program that promotes EMT and metastasis. By contrast, molecular glues and PROTACs are being developed to directly degrade EF. Collectively, our findings underscore the necessity of completely eliminating EF activity, since incomplete inhibition may paradoxically promote metastatic progression and relapse. At the same time, we emphasize that additional validation across diverse EwS models will be required to distinguish broadly conserved from context-dependent EF-regulated pathways.”

Concern: *have the authors been able to study sequential samples in patients, in terms of expression profiles, and EF dosages at different time point of disease course.*

Our response: Unfortunately, we have access to retrospective (Affymetrix) gene expression data for only a very small number of paired primary and metastatic tumor samples, and it is unclear whether these represent true sequential samples from the same patients over time. Thus, a definitive analysis of EF dosage dynamics across disease progression in patients is currently not feasible. Nevertheless, our results support and refine previous speculations (Franzetti *et al.*, Seong *et al.*, Aynaud *et al.*), which suggest that EF-low and fluctuating metastable cell states are transiently required for EMT and tumor cell dissemination, whereas optimal EF expression levels favor metastatic outgrowth at distant sites. Accordingly, we do not

expect significant differences in EF expression levels or transcriptional activity when comparing metastatic and primary patient samples broadly. This interpretation aligns with previous findings (Aynaud *et al*, 2020; Franzetti, 2017) which demonstrated that EF-low cell fractions represent only a minor (1-2%) subset in patient biopsy samples and that such states are reversible in cell line models, supporting the transient nature of EF fluctuations in vivo.

Concern: *have the authors tested the impact of further overexpression of EF in A673 cells in terms of phenotype and gene expression.*

Our response: Our system precisely modulates endogenous EF but does not increase EF beyond physiological levels. Overexpression would require ectopically forced EF expression, which falls outside our focus on endogenous dosage dynamics. It has been performed by others in multiple Ewing sarcoma cell lines including A673 resulting in growth suppression and apoptosis (Seong *et al*, 2021).

Referee #2:

I strongly believe that plasticity in the expression of sarcoma-driving fusion oncogenes is a major factor in the malignancy of these cancers and has not received sufficient attention thus far. Any project addressing these issues comes with major challenges. I applaud the authors for embarking on this journey and urge the editorial team to consider the general importance of this topic.

We appreciate the reviewer's thoughtful evaluation of EF plasticity as a key driver of EwS malignancy and thank them for highlighting the broader importance of this topic.

Concern: *Obviously, the major weakness of this paper is that all experiments were carried out in one high-passage cell line. The authors evaluated two clones and executed careful control experiments. Nevertheless, the fact that they were not able to reproduce their results in another line is somewhat concerning and puts the general applicability of their findings in question. Are the conclusions from this study truly representative for Ewing sarcomas? This limitation should be stated very clearly in the manuscript.*

Our response: We agree that this is a central limitation of our study. As we were unable to establish the homozygous endogenous degron knock-in in other EwS cell lines (see new Appendix Fig. S1A), we could not directly reproduce our results in a different genetic background. However, despite the restriction of our model to A673 cells, the EF dosage dependent phenotypic changes in cell adherence and migration have previously been reported in SK-N-MC (Franzetti *et al*, 2017) and validated in TC71 and EWS502 EwS cells (Chaturvedi *et al*, 2012). Additionally, using independent perturbations, the role of EF in driving EMT, dormancy and stemness has been studied in EW-8, ES6 and ES8 cells (Khoogar *et al*, 2022), supporting the general relevance of EF-dosage driven plasticity across EwS contexts. While our study provides proof-of-principle that EF dosage modulates discrete transcriptional programs and migratory/invasive phenotypes, there may be inter-individual variation in EF-dose sensitivity and fluctuations across genetic backgrounds. This is now clearly stated in the Discussion, lines 292-297:

“Of note, despite the restriction of our model to A673 cells, the EF linked phenotypic changes in cell adherence and migration have previously been reported in SK-N-MC (Franzetti *et al.*, 2017) and validated in TC71 and EWS502 EwS cells (Chaturvedi *et al.*, 2012). Additionally, using independent perturbations, the role of EF in driving EMT, dormancy and stemness has been studied in EW-8, ES6 and ES8 cells (Khoogar *et al.*, 2022), supporting the general relevance of EF-driven plasticity across EwS contexts.”

and 355-358.

“While our study in A673 cells provides proof-of-principle that EF dosage modulates discrete transcriptional programs and migratory/invasive phenotypes, there may be inter-individual variation in EF-dose sensitivity and fluctuations across genetic backgrounds.”

Concern: *The study includes extensive bioinformatic analyses to support the notion that enhancers containing GGAA microsatellites and Krüppel-like zinc finger transcription factors act as co-regulators of genes with differential expression in response to EWS::FLI1 dose. Experimental evidence to validate the functional relevance of these observations would strengthen the manuscript substantially.*

Our response: GGAA microsatellites as EF response elements in EwS are well documented in the literature (Gangwal *et al.*, 2008; Guillon *et al.*, 2009; Kinsey *et al.*, 2006; Luo *et al.*, 2009). Our study contributes further by showing EF dose-dependency and genomic context of these elements. We agree with this reviewer that functional validation of predicted EF transcriptional co-regulators, including Krüppel-like zinc finger transcription factors, is a logical next step in follow-up of our study. However, in keeping with the editor’s explicit recommendation, we consider these additional experimental studies beyond the scope of this communication.

We now mention this in the Discussion (line 340-342) as follows:

“So far, nothing is known about functional interactions between EF and any of these candidate EF-interacting transcription factors, and experimental validation is warranted in follow-up studies.”

However, we have utilized publicly available ChIP-seq data (Shi *et al.*, 2020) for one of our top Krüppel-like zinc finger transcription factor candidates, KLF15, in A673 cells. We found that ~35% peaks (3616/10458) overlap with FLI1 binding sites, supporting the potential for coregulator validation in future studies.

Concern: *Another relevant weakness is that previous studies indicated that differences in the expression of EWS::FLI1 were dynamic over time and truly plastic. How do the authors reconcile these published findings with persistent dysregulation of certain genes after transient reduction of EWS::FLI1 dose? The authors need to discuss the discrepancy of their findings with published literature.*

Our response: We reconcile published with our findings by time and amplitude dependence of EF modulation. In contrast to sustained transcriptomic changes, persistent phenotypic changes were observed only after recovery from a 21-day modulation period, not after 7 days (Fig. 6A and EV5A). Moreover, persistent transcriptomic changes after EF recovery affected only a

limited subset (106 genes staying up, 17 genes staying down) rather than the entire EF program. This indicates partial, state-dependent memory rather than global irreversibility.

Prior works (Franzetti *et al.*, 2017; Wrenn *et al.*, 2023) using cell surface markers such as ICAM1 and CD73 to define presence of EF-low or cancer-associated fibroblasts-like tumor cell populations in the A673 cell line documented re-equilibration of EF heterogeneity after 7 days (CD73) or 20 days (ICAM1).

We reproduced these findings in our model after flow-sorting of EF-mNG^{Dim} and EF-mNG^{High} cells. As shown in Reviewer Fig. 1, sorted EF-mNG^{Dim} and EF-mNG^{High} cells regenerated heterogeneity starting at 112 and 72 hours, respectively. This confirms the dynamic nature of EF expression levels and high plasticity of EwS subpopulations.

Reviewer Figure 1: Dynamic nature of EF expression in A2.2 dTAG cells.

(A) Sorting strategy of A2.2 dTAG clone into subpopulations with low (mNG^{Dim}) and high (mNG^{High}) EF-mNG fluorescence intensity.

(B) Quantification of average green object mean intensity (Incucyte S3-C2) of sorted subpopulations normalized to the heterogeneous unsorted A2.2 population, starting 12 h after sorting. Dashed lines denote the equilibrium threshold ($\pm 10\%$), used to define the timepoint at which sorted populations converge with the heterogeneous state and remain

within this range for at least three consecutive measurements. Red circle indicates the equilibrium timepoint of EF-mNG^{High} cells, blue circle indicates that of EF-mNG^{Dim} cells.

(C) Flow cytometry analysis of mNG fluorescence intensity in the sorted and heterogeneous populations at day 3 and day 6.

Thus, our data refine earlier findings and imply that both duration and magnitude of EF fluctuation, whether natural or therapy-induced, determine whether cells revert or transition into a metastable invasive state. This is now stated in lines 368-372 as follows:

“Importantly, these persistent changes did not involve the entire EF program but instead a limited subset of genes, consistent with a state-dependent “memory” rather than global irreversibility. Together with prior studies showing rapid re-equilibration of EF heterogeneity at the population level (Franzetti et al., 2017; Wrenn et al., 2025), our results refine earlier models and suggest that both the duration and magnitude of EF fluctuation determine whether cells revert or transition into a metastable invasive state.”

Importantly, related findings by Franzetti et al. confirmed increased incidence of lung nodule formation without *in vivo* sustained EF knockdown consistent with our observation that transient EF modulation (21 days 15 nM dTAG^v-1 treatment) can prime metastatic behavior even after EF recovery (Fig. 6E-F).

Concern: *The authors show that expression of 23 out of 85 genes (associated with EMT and reduced EWS::FLI1 dose) correlated with survival. I am sorry, but I think this correlation is fairly random and does not enhance the relevance of the main findings reported in this paper. I would consider removing figure 7 from the manuscript.*

Our response: Following the reviewer’s and the editor’s recommendation, we removed Figure 7 from the revised manuscript. Accordingly, we have changed the manuscript title to exclude the potential prognostic aspect of our study. It now reads:

“Modelling EWS::FLI1 protein fluctuations reveal determinants of tumor plasticity in Ewing sarcoma”

Concern: *In addition, I would like to recommend further discussion of the concept of plasticity. The authors emphasize that therapies aimed at eliminating EWS::FLI1 need to completely eradicate its activity and must avoid pro-tumorigenic effects at lower EWS::FLI1 doses. However, EWS::FLI1 plasticity may have important implications that go far beyond this statement in terms of development of resistance and metronomic therapy. This should be discussed.*

Our response: We thank all three Reviewers for this recommendation. We have significantly expanded the final paragraph (lines 402-419) of the Discussion to address broader therapeutic implications of EF plasticity, including resistance and metronomic therapy as follows:

“The therapeutic ramifications of our findings are multifaceted. EwS is particularly sensitive to DNA-damaging agents due to EF-mediated defects in DNA double-strand break repair (Gorthi et al., 2018; Lee et al., 2020; Menon et al, 2025), and reduced EF expression may decrease responsiveness to such therapies. Metronomic treatment strategies, the continuous administration of low-dose anticancer agents currently under consideration for EwS maintenance therapy, may also be negatively affected by fluctuating EF levels, as drugs that reduce EF

St. Anna Kinderkrebsforschung

CHILDREN'S CANCER RESEARCH INSTITUTE

activity could unintentionally induce invasive, metastasis-prone hybrid states when applied at low dose. For drugs that modulate EF DNA-binding or transcriptional activity, such as cytarabine (Stegmaier et al, 2007), YK-4-279 and TK216 (Erkizan et al, 2009; Meyers et al, 2024), trabectedin (Grohar et al, 2011a), mithramycin and other mitralogues (Grohar et al, 2011b; Osgood et al, 2016), midostaurin (Boro et al., 2012), low-dose actinomycin (Chen et al, 2013b), shikonin (Chen et al, 2013a), and HCI2509 (Sankar et al., 2014) our results emphasize the need to define EF dose-dependent gene sets and phenotypes, since suboptimal inhibition may foster pro-metastatic states. Most of these compounds have been demonstrated to inhibit GGAA microsatellite-driven EF target gene activation rather than directly targeting EF protein levels, and little is known about their impact on the EF-repressive program that promotes EMT and metastasis. By contrast, molecular glues and PROTACs are being developed to directly degrade EF. Collectively, our findings underscore the necessity of completely eliminating EF activity, since incomplete inhibition may paradoxically promote metastatic progression and relapse. At the same time, we emphasize that additional validation across diverse EwS models will be required to distinguish broadly conserved from context-dependent EF-regulated pathways.”

If the authors can address the issues outlined above and clearly discuss the limitations of the study, I recommend to consider accepting this paper in light of the fact that it sheds further light on complex and underappreciated aspect of Ewing sarcoma biology.

We hope that the revised version of our manuscript sufficiently discusses the limitations of our study.

Referee #3:

The manuscript seeks to examine the effect of fluctuation in the level of EWS/FLI (EF) fusion oncogene, which are known to arise in EWS and to promote metastatic activity of EWS. Using an elegant strategy to edit the endogenous fusion gene with a degron c-terminal tag the authors undertake high end transcriptome characterisation and alongside phenotypic response monitoring in Ewing sarcoma A567 cells.

Experiments are very well designed and controlled and data presented are convincing. Overall, the manuscript confirms existing evidence concerning the response of EWS oncogenic signalling to changes in the dose of EF fusion oncogene product and extends knowledge of the transcriptional regulatory landscape involved in process. Amongst others the work raises novel hypothesis concerning the involvement of cellular transcription factors that may deliver EF signalling output.

We thank this reviewer for the overall positive perception of our study and for highlighting important limitations that require clarification.

Concern: *The study has several limitations that should be clearly acknowledged. First, it relies on the use of a single Ewing sarcoma (EWS) cell line (A567), and functional gene editing could not be achieved in other EWS cell lines. This restricts the generalizability of the findings and raises*

questions about whether the observed regulatory networks are broadly relevant across EWS subtypes

Our response: We fully agree that this is a limitation. Our degron-tagging model could only be established in A673, but phenotypic consequences of EF reduction have independently been reported using RNAi-mediated knockdown in SK-N-MC (Franzetti *et al.*, 2017), validated in TC71 and EWS502 Ewing sarcoma cells with respect to cell adherence and migration (Chaturvedi *et al.*, 2012), and extended to EW-8, ES6 and ES8 cells with respect to EMT, dormancy and stemness (Khoogar *et al.*, 2022). These concordant findings strongly suggest the phenotypes are not unique to A673, although sensitivity to EF dosage may vary. Nonetheless, our transcriptomic dissection of EF dosage sensitivity was performed in a single background. We therefore frame our study as a proof-of-principle and recognize that the quantitative details of EF-regulatory networks and their phenotypic consequences may vary between EwS contexts. To prove the generalizability of the gene regulatory networks identified in A673 cells, functional validation experiments in other cell lines will be required (as now mentioned in line 378). However, in keeping with the editor's recommendation, we consider these experimental follow-up studies beyond the scope of our current communication.

We now explicitly acknowledge this limitation in the Discussion section, lines 292-297:

*"Of note, despite the restriction of our model to A673 cells, the EF linked phenotypic changes in cell adherence and migration have previously been reported in SK-N-MC (Franzetti *et al.*, 2017) and validated in TC71 and EWS502 EwS cells (Chaturvedi *et al.*, 2012). Additionally, using independent perturbations, the role of EF in driving EMT, dormancy and stemness has been studied in EW-8, ES6 and ES8 cells (Khoogar *et al.*, 2022), supporting the general relevance of EF-driven plasticity across EwS contexts."*

and 355-358.

"While our study in A673 cells provides proof-of-principle that EF dosage modulates discrete transcriptional programs and migratory/invasive phenotypes, there may be inter-individual variation in EF-dose sensitivity and fluctuations across genetic backgrounds."

Concern: *Moreover, the inability to edit EF in other cell lines suggests that certain molecular functions of EF may be essential in those contexts but are dispensable in A567. This functional variability limits the ability to model and safely predict the general or principal impact of EF-targeting therapeutics. In other words, potential EF functions that are not evident in A567 may counteract or overshadow the risk-such as enhanced metastasis-identified in this study. Therefore, any claims regarding EF-targeted therapies should be moderated and carefully weighed against this context. Finally, the conclusions drawn in the study are largely based on correlative data and have not been experimentally validated. Additional functional validation will be essential in future work. These limitations should be more explicitly discussed in the manuscript's discussion section to provide a balanced and accurate interpretation of the findings.*

Response: We agree with this reviewer that tolerance to C-terminal tagging in A673 but not other lines suggests that certain EF functions may indeed be indispensable in specific

genetic/epigenetic contexts. This underscores the challenge of developing EF perturbation models and the need for caution when extrapolating therapeutic implications indicating the need for future experimental validation studies. A recent preprint (McGinnis *et al*, 2024) reported successful targeting of endogenous *EWSR1::FLI1* with an inducible SMASH-degron tag, which might be used for validation of our results in other EwS cell lines in the future. We have carefully revised the discussion to present our conclusions in a more balanced way (lines 277-281).

“However, our unsuccessful attempts to establish a similar model in other EwS cell lines suggest that tolerance to carboxy-terminal EF modification may depend on the underlying genomic or epigenomic background reflecting a cellular adaptive response to maintain EF function, likely shaped by each line’s unique (epi-)genetic context. The precise molecular mechanisms for this selectivity remain to be determined. Therefore, the restriction to a single cell line has to be seen as the major limitation of our study.”

The paragraph addressing potential consequences for Ewing sarcoma therapy has been extensively modified to address all three reviewers’ critiques as follows (lines 402-419):

“The therapeutic ramifications of our findings are multifaceted. EwS is particularly sensitive to DNA-damaging agents due to EF-mediated defects in DNA double-strand break repair (Gorthi *et al.*, 2018; Lee *et al.*, 2020; Menon *et al*, 2025), and reduced EF expression may decrease responsiveness to such therapies. Metronomic treatment strategies, the continuous administration of low-dose anticancer agents currently under consideration for EwS maintenance therapy, may also be negatively affected by fluctuating EF levels, as drugs that reduce EF activity could unintentionally induce invasive, metastasis-prone hybrid states when applied at low dose. For drugs that modulate EF DNA-binding or transcriptional activity, such as cytarabine (Stegmaier *et al*, 2007), YK-4-279 and TK216 (Erkizan *et al*, 2009; Meyers *et al*, 2024), trabectedin (Grohar *et al*, 2011a), mithramycin and other mitralogues (Grohar *et al*, 2011b; Osgood *et al*, 2016), midostaurin (Boro *et al.*, 2012), low-dose actinomycin (Chen *et al*, 2013b), shikonin (Chen *et al*, 2013a), and HCI2509 (Sankar *et al.*, 2014) our results emphasize the need to define EF dose-dependent gene sets and phenotypes, since suboptimal inhibition may foster pro-metastatic states. Most of these compounds have been demonstrated to inhibit GGAA microsatellite-driven EF target gene activation rather than directly targeting EF protein levels, and little is known about their impact on the EF-repressive program that promotes EMT and metastasis. By contrast, molecular glues and PROTACs are being developed to directly degrade EF. Collectively, our findings underscore the necessity of completely eliminating EF activity, since incomplete inhibition may paradoxically promote metastatic progression and relapse. At the same time, we emphasize that additional validation across diverse EwS models will be required to distinguish broadly conserved from context-dependent EF-regulated pathways.”

Response references:

Aguirre AJ, Meyers RM, Weir BA, Vazquez F, Zhang C-Z, Ben-David U, Cook A, Ha G, Harrington WF, Doshi MB *et al* (2016) Genomic Copy Number Dictates a Gene-Independent Cell Response to CRISPR/Cas9 Targeting. *Cancer Discovery* 6: 914-929

Aynaud MM, Mirabeau O, Gruel N, Grossetete S, Boeva V, Durand S, Surdez D, Saulnier O, Zaidi S, Gribkova S *et al* (2020) Transcriptional Programs Define Intratumoral Heterogeneity of Ewing Sarcoma at Single-Cell Resolution. *Cell Rep* 30: 1767-1779 e1766

Biddle A, Gammon L, Liang X, Costea DE, Mackenzie IC (2016) Phenotypic Plasticity Determines Cancer Stem Cell Therapeutic Resistance in Oral Squamous Cell Carcinoma. *EBioMedicine* 4: 138-145

Boro A, Pretre K, Rechfeld F, Thalhammer V, Oesch S, Wachtel M, Schafer BW, Niggli FK (2012) Small-molecule screen identifies modulators of EWS/FLI1 target gene expression and cell survival in Ewing's sarcoma. *Int J Cancer* 131: 2153-2164

Chaturvedi A, Hoffman LM, Welm AL, Lessnick SL, Beckerle MC (2012) The EWS/FLI Oncogene Drives Changes in Cellular Morphology, Adhesion, and Migration in Ewing Sarcoma. *Genes Cancer* 3: 102-116

Chen C, Shanmugasundaram K, Rigby AC, Kung AL (2013a) Shikonin, a natural product from the root of *Lithospermum erythrorhizon*, is a cytotoxic DNA-binding agent. *Eur J Pharm Sci* 49: 18-26

Chen C, Wonsey DR, Lemieux ME, Kung AL (2013b) Differential disruption of EWS-FLI1 binding by DNA-binding agents. *PLoS One* 8: e69714

Erkizan HV, Kong Y, Merchant M, Schlottmann S, Barber-Rotenberg JS, Yuan L, Abaan OD, Chou TH, Dakshanamurthy S, Brown ML *et al* (2009) A small molecule blocking oncogenic protein EWS-FLI1 interaction with RNA helicase A inhibits growth of Ewing's sarcoma. *Nat Med* 15: 750-756

Franzetti GA, Laud-Duval K, van der Ent W, Brisac A, Irondelle M, Aubert S, Dirksen U, Bouvier C, de Pinieux G, Snaar-Jagalska E *et al* (2017) Cell-to-cell heterogeneity of EWSR1-FLI1 activity determines proliferation/migration choices in Ewing sarcoma cells. *Oncogene* 36: 3505-3514

Fustaino V, Presutti D, Colombo T, Cardinali B, Papoff G, Brandi R, Bertolazzi P, Felici G, Ruberti G (2017) Characterization of epithelial-mesenchymal transition intermediate/hybrid phenotypes associated to resistance to EGFR inhibitors in non-small cell lung cancer cell lines. *Oncotarget; Vol 8, No 61*

Gangwal K, Sankar S, Hollenhorst PC, Kinsey M, Haroldsen SC, Shah AA, Boucher KM, Watkins WS, Jorde LB, Graves BJ *et al* (2008) Microsatellites as EWS/FLI response elements in Ewing's sarcoma. *Proc Natl Acad Sci U S A* 105: 10149-10154

Gorthi A, Romero JC, Loranc E, Cao L, Lawrence LA, Goodale E, Iniguez AB, Bernard X, Masamsetti VP, Roston S *et al* (2018) EWS-FLI1 increases transcription to cause R-loops and block BRCA1 repair in Ewing sarcoma. *Nature* 555: 387-391

Grohar PJ, Griffin LB, Yeung C, Chen QR, Pommier Y, Khanna C, Khan J, Helman LJ (2011a) Ecteinascidin 743 interferes with the activity of EWS-FLI1 in Ewing sarcoma cells. *Neoplasia* 13: 145-153

Grohar PJ, Woldemichael GM, Griffin LB, Mendoza A, Chen QR, Yeung C, Currier DG, Davis S, Khanna C, Khan J *et al* (2011b) Identification of an inhibitor of the EWS-FLI1 oncogenic transcription factor by high-throughput screening. *J Natl Cancer Inst* 103: 962-978

Guillon N, Tirode F, Boeva V, Zynovyev A, Barillot E, Delattre O (2009) The oncogenic EWS-FLI1 protein binds in vivo GGAA microsatellite sequences with potential transcriptional activation function. *PLoS One* 4: e4932

Hanrahan K, O'Neill A, Prencipe M, Bugler J, Murphy L, Fabre A, Puhr M, Culig Z, Murphy K, Watson RW (2017) The role of epithelial–mesenchymal transition drivers ZEB1 and ZEB2 in mediating docetaxel-resistant prostate cancer. *Molecular Oncology* 11: 251-265

Khoogar R, Li F, Chen Y, Ignatius M, Lawlor ER, Kitagawa K, Huang TH, Phelps DA, Houghton PJ (2022) Single-cell RNA profiling identifies diverse cellular responses to EWSR1/FLI1 downregulation in Ewing sarcoma cells. *Cell Oncol (Dordr)* 45: 19-40

Kinsey M, Smith R, Lessnick SL (2006) NR0B1 Is Required for the Oncogenic Phenotype Mediated by EWS/FLI in Ewing's Sarcoma. *Molecular Cancer Research* 4: 851-859

Lee Sg, Kim N, Kim Sm, Park IB, Kim H, Kim S, Kim Bg, Hwang JM, Baek IJ, Gartner A *et al* (2020) Ewing sarcoma protein promotes dissociation of poly(ADP-ribose) polymerase 1 from chromatin. *EMBO reports* 21: e48676

Luo W, Gangwal K, Sankar S, Boucher KM, Thomas D, Lessnick SL (2009) GSTM4 is a microsatellite-containing EWS/FLI target involved in Ewing's sarcoma oncogenesis and therapeutic resistance. *Oncogene* 28: 4126-4132

McGinnis JH, Enriquez AB, Vandiver F, Bai X, Kim J, Kilgore J, Saha P, O'Hara R, Xie Y, Banaszynski LA *et al* (2024) Endogenous EWSR1-FLI1 degraon alleles enable control of fusion oncoprotein expression in tumor cell lines and xenografts. *bioRxiv*: 2024.2010.2027.620498

Menon S, Gracilla D, Breese MR, Lin YP, Dela Cruz F, Feinberg T, de Stanchina E, Galic A-F, Allegakoen H, Perati S *et al* (2025) FET fusion oncoproteins disrupt physiologic DNA repair and create a targetable opportunity for ATR inhibitor therapy. *bioRxiv*: 2023.2004.2030.538578

Meyers PA, Federman N, Daw N, Anderson PM, Davis LE, Kim A, Macy ME, Pietrofeso A, Ratan R, Riedel RF *et al* (2024) Open-Label, Multicenter, Phase I/II, First-in-Human Trial of TK216: A First-Generation EWS::FLI1 Fusion Protein Antagonist in Ewing Sarcoma. *J Clin Oncol* 42: 3725-3734

Osgood CL, Maloney N, Kidd CG, Kitchen-Goosen S, Segars L, Gebregiorgis M, Woldemichael GM, He M, Sankar S, Lessnick SL *et al* (2016) Identification of Mithramycin Analogues with Improved Targeting of the EWS-FLI1 Transcription Factor. *Clin Cancer Res* 22: 4105-4118

Sankar S, Theisen ER, Bearss J, Mulvihill T, Hoffman LM, Sorna V, Beckerle MC, Sharma S, Lessnick SL (2014) Reversible LSD1 inhibition interferes with global EWS/ETS transcriptional activity and impedes Ewing sarcoma tumor growth. *Clin Cancer Res* 20: 4584-4597

Seong BKA, Dharia NV, Lin S, Donovan KA, Chong S, Robichaud A, Conway A, Hamze A, Ross L, Alexe G *et al* (2021) TRIM8 modulates the EWS/FLI oncoprotein to promote survival in Ewing sarcoma. *Cancer Cell* 39: 1262-1278 e1267

Shi X, Zheng Y, Jiang L, Zhou B, Yang W, Li L, Ding L, Huang M, Gery S, Lin DC *et al* (2020) EWS-FLI1 regulates and cooperates with core regulatory circuitry in Ewing sarcoma. *Nucleic Acids Res* 48: 11434-11451

Stegmaier K, Wong JS, Ross KN, Chow KT, Peck D, Wright RD, Lessnick SL, Kung AL, Golub TR (2007) Signature-based small molecule screening identifies cytosine arabinoside as an EWS/FLI modulator in Ewing sarcoma. *PLoS Med* 4: e122

St. Anna Kinderkrebsforschung

CHILDREN'S CANCER RESEARCH INSTITUTE

Wei X, Ge Y, Zheng Y, Zhao S, Zhou Y, Chang Y, Wang N, Wang X, Zhang J, Zhang X *et al* (2025) Hybrid EMT Phenotype and Cell Membrane Tension Promote Colorectal Cancer Resistance to Ferroptosis. *Advanced Science* 12: 2413882

Wrenn ED, Apfelbaum AA, Rudzinski ER, Deng X, Jiang W, Sud S, Van Noord RA, Newman EA, Garcia NM, Miyaki A *et al* (2023) Cancer-Associated Fibroblast-Like Tumor Cells Remodel the Ewing Sarcoma Tumor Microenvironment. *Clin Cancer Res* 29: 5140-5154

Yu M, Bardia A, Wittner BS, Stott SL, Smas ME, Ting DT, Isakoff SJ, Ciciliano JC, Wells MN, Shah AM *et al* (2013) Circulating Breast Tumor Cells Exhibit Dynamic Changes in Epithelial and Mesenchymal Composition. *Science* 339: 580-584

Heinrich Kovar, PhD

3rd Nov 2025

Dear Prof. Kovar,

Thank you for submitting your revised study. We have now received the reports from referees #1 and #2. As you will see below, they are satisfied with the revisions, and I will therefore be able to accept your manuscript once the following editorial concerns are addressed:

1/ Manuscript text:

- Please remove the yellow highlights and only indicate in track changes mode any new modification in the text.
- Methods:
 - o Please add the suppl. Methods to the Methods section to the main manuscript text.
 - o Statistics: please provide a statement on randomization and fill in the corresponding section in the author checklist.
 - o Remove bioRender from the acknowledgments and add a dedicated section to the Methods using this format:

Graphics:

(some of the... OR Figure #... OR synopsis) Graphics were created with BioRender.com.

- Disclosure and competing interest statement: please add that Karla Queiroz and Dorota Kurek work for Mimetas, a biotech company in The Netherlands (see our policy at <https://www.embopress.org/competing-interests>).

2/ Figures:

- Please consider adding reviewer figure 1 in the manuscript, as suggested by ref #2.
- Carefully check the composition of Figure 2B, Figure EV1, Figure EV5. Please note that re-use of figure panels is allowed but should be mentioned in the legend.
- The figure quality of Appendix Figure S1 is very low; please supply a high resolution figure.
- We can now accommodate more than 5 EV figures, and you are welcome to make your Appendix figures EV figures. If you choose to keep an Appendix, please correct the nomenclature for the figures to "Appendix Figure S1" -S3. Additional separate figure files are not needed.
- Dataset EV tables: there is no need to zip the files, they can be uploaded in excel format and as file type dataset.
- Please address the queries from our data editors in the figure legends:
 1. Please indicate the statistical test used for data analysis in the legend of figure EV1 H
 2. Please note that the box plots need to be defined in terms of minima, maxima, centre, bounds of box and whiskers, and percentile in the legends of figures S3 D

3/ Please note that all corresponding authors are required to supply an ORCID ID for their name upon submission of a revised manuscript. An ORCID identifier is currently missing for Valerie Fock.

4/ I introduced minor edits in The paper explained, please let me know if you agree or amend as you see fit:

Problem

Ewing sarcoma (EwS) is a highly aggressive bone cancer driven by the fusion oncoprotein EWS::FLI1 (EF). While current therapies improve outcomes for localized disease, outcomes for patients with metastatic or relapsed tumors remain poor. Despite EF's central role in tumorigenesis, the influence of fluctuations in its expression on pro-metastatic cell plasticity remains unclear, limiting the design of effective targeted strategies.

Results

Using a tunable EF expression system, we demonstrated that even transient and modest reductions in EF levels can promote a persistent migratory and invasive tumor cell state. Transcriptomic profiling revealed that EF controls distinct, dosage-dependent gene expression programs. By analyzing the chromatin architecture at EF-bound regions, we identified cofactors that cooperate with EF to regulate these programs. Importantly, we demonstrate that transient fluctuations in EF levels result in the sustained activation of genes associated with epithelial-mesenchymal plasticity.

Impact

This study highlights the pathobiological impact of dynamic fluctuations in oncoprotein levels on disease progression in cancers with otherwise inactive genomes, such as EwS. It suggests that therapeutic strategies targeting driver oncoproteins must completely eradicate their activity to prevent inadvertent promotion of metastasis. Furthermore, the study provides a valuable resource of genes sensitive to EF dosage and candidate co-regulators, furthering our understanding of EwS biology and revealing new therapeutic targets.

5/ Synopsis:

Thank you for providing a nice visual abstract. Unfortunately, when resized to 550 px wide by 300-600 px high, the text becomes illegible. Could you please resize it and increase the font size to make it more legible? I have cropped a small portion of the image to use as a thumbnail for the table of contents on our website (see attached). Please let me know if you agree, or if you would prefer to provide an alternative image with the same dimensions (115px x 70px).

I have added minor edits to your synopsis text, please let me know if you agree or amend as you see fit:

The partial and transient suppression of the EF oncoprotein was found to paradoxically promote Ewing sarcoma metastatic potential through persistent transcriptional dysregulation, establishing the oncoprotein dosage dynamics as a key regulator of metastatic plasticity and therapeutic vulnerability.

- Graded EF depletion using dTAGV-1 elicited a non-linear phenotypic response, with modest EF loss being sufficient to induce a pro-metastatic state.
- Integration of acute transcriptional responses with chromatin profiling led to the identification of EF co-regulators.
- EF restoration following transient depletion resulted in lasting dysregulation of pro-metastasis-related genes and emergence of hybrid epithelial-mesenchymal states, thereby enhancing extravasation and lung metastasis in vivo.
- Partial or transient EF inhibition may promote disease progression and worsen clinical outcomes, highlighting that therapeutic strategies targeting EF must achieve complete and sustained suppression.

6/ As part of the EMBO Publications transparent editorial process initiative (see our Editorial at <http://embomolmed.embopress.org/content/2/9/329>), EMBO Molecular Medicine will publish online a Review Process File (RPF) to accompany accepted manuscripts.

This file will be published in conjunction with your paper and will include the anonymous referee reports, your point-by-point response and all pertinent correspondence relating to the manuscript. Let us know whether you agree with the publication of the RPF and as here, if you want to remove or not any figures from it prior to publication.

I look forward to receiving your revised manuscript.

Yours sincerely,

Lise Roth

***** Reviewer's comments *****

Referee #1 (Comments on Novelty/Model System for Author):

Investigate in the clinics

Referee #1 (Remarks for Author):

The authors have adequately addressed the questions of the reviewers

Referee #2 (Remarks for Author):

The authors considered my comments and the other reviewers' comments very carefully. The manuscript was expanded in response to the concerns brought up during the first review. The changes fully address my concerns.

I am a bit surprised, that I cannot find reviewer figure 1 in the revised manuscript. Did I miss it? Personally, I recommend to include it in the appendix.

I very much appreciate the thoughtful revision by the authors and wish to congratulate them on improving the manuscript substantially.

The authors addressed the remaining editorial issues.

18th Nov 2025

Dear Prof. Kovar,

Thank you for submitting your revised files and dealing with the remaining editorial issues. I am pleased to inform you that your manuscript is accepted for publication and is now being sent to our publisher to be included in the next available issue of EMBO Molecular Medicine.

Please note that I have removed the following sentence from the acknowledgements:

"For open access purposes, the authors have applied a CC BY public copyright license to any author-accepted manuscript version arising from this submission." as all papers submitted are published Open Access (OA) under a Creative Commons CC-BY 4.0 license.

Yours sincerely,

Lise Roth
